# Assembly of recombinant tau into filaments identical to those of Alzheimer's disease and chronic traumatic encephalopathy

Sofia Lövestam[1], Fujiet Adrian Koh[2], Bart van Knippenberg[2], Abhay Kotecha[2], Alexey G Murzin[1], Michel Goedert[1]*, Sjors HW Scheres[1]*

[1]Medical Research Council Laboratory of Molecular Biology, Cambridge, United Kingdom; [2]Thermo Fisher Scientific, Eindhoven, Netherlands

**Abstract** Abundant filamentous inclusions of tau are characteristic of more than 20 neurodegenerative diseases that are collectively termed tauopathies. Electron cryo-microscopy (cryo-EM) structures of tau amyloid filaments from human brain revealed that distinct tau folds characterise many different diseases. A lack of laboratory-based model systems to generate these structures has hampered efforts to uncover the molecular mechanisms that underlie tauopathies. Here, we report in vitro assembly conditions with recombinant tau that replicate the structures of filaments from both Alzheimer's disease (AD) and chronic traumatic encephalopathy (CTE), as determined by cryo-EM. Our results suggest that post-translational modifications of tau modulate filament assembly, and that previously observed additional densities in AD and CTE filaments may arise from the presence of inorganic salts, like phosphates and sodium chloride. In vitro assembly of tau into disease-relevant filaments will facilitate studies to determine their roles in different diseases, as well as the development of compounds that specifically bind to these structures or prevent their formation.

*For correspondence:
mg@mrc-lmb.cam.ac.uk (MG);
scheres@mrc-lmb.cam.ac.uk
(SHWS)

## Editor's evaluation

In this paper 76 cryo-EM structures of recombinant tau filaments assembled in vitro are described. This is a scientific tour-de-force, and will provide an immense database that can be used by everyone working in the amyloid field. When this knowledge is combined with the structure of tau filaments in vivo, it will help shape the design of laboratory experiments in the future to generate the conditions to replicate the in vivo forms in vitro.

## Introduction

Six tau isoforms that range from 352 to 441 amino acids in length are expressed in adult human brain from a single gene by alternative mRNA splicing (*Goedert et al., 1989*). The tau sequence can be separated into an N-terminal projection domain (residues 1–150), a proline-rich region (residues 151–243), the microtubule-binding repeat region (residues 244–368), and a C-terminal domain (residues 369–441). Isoforms differ by the presence or absence of 1 or 2 inserts of 29 amino acids each, in the N-terminal region (0N, 1N or 2N), and the presence or absence of the second of four microtubule-binding repeats of 31 or 32 amino acids each (resulting in three-repeat, 3R, and four-repeat, 4R, isoforms). The second N-terminal insert is only expressed together with the first. Tau filaments can differ in isoform composition in disease (*Goedert et al., 2017*). Thus, a mixture of all six isoforms is present in the tau filaments of Alzheimer's disease (AD), chronic traumatic encephalopathy (CTE), and

**eLife digest** Many neurodegenerative diseases, including Alzheimer's disease, the most common form of dementia, are characterised by knotted clumps of a protein called tau. In these diseases, tau misfolds, stacks together and forms abnormal filaments, which have a structured core and fuzzy coat. These sticky, misfolded proteins are thought to be toxic to brain cells, the loss of which ultimately causes problems with how people move, think, feel or behave.

Reconstructing the shape of tau filaments using an atomic-level imaging technique called electron cryo-microscopy, or cryo-EM, researchers have found distinct types of tau filaments present in certain diseases. In Alzheimer's disease, for example, a mixture of paired helical and straight filaments is found. Different tau filaments are seen again in chronic traumatic encephalopathy (CTE), a condition associated with repetitive brain trauma.

It remains unclear, however, how tau folds into these distinct shapes and under what conditions it forms certain types of filaments. The role that distinct tau folds play in different diseases is also poorly understood. This is largely because researchers making tau proteins in the lab have yet to replicate the exact structure of tau filaments found in diseased brain tissue.

Lövestam et al. describe the conditions for making tau filaments in the lab identical to those isolated from the brains of people who died from Alzheimer's disease and CTE. Lövestam et al. instructed bacteria to make tau protein, optimised filament assembly conditions, including shaking time and speed, and found that *bona fide* filaments formed from shortened versions of tau. On cryo-EM imaging, the lab-produced filaments had the same left-handed twist and helical symmetry as filaments characteristic of Alzheimer's disease.

Adding salts, however, changed the shape of tau filaments. In the presence of sodium chloride, otherwise known as kitchen salt, tau formed filaments with a filled cavity at the core, identical to tau filaments observed in CTE. Again, this structure was confirmed on cryo-EM imaging.

Being able to make tau filaments identical to those found in human tauopathies will allow scientists to study how these filaments form and elucidate what role they play in disease. Ultimately, a better understanding of tau filament formation could lead to improved diagnostics and treatments for neurodegenerative diseases involving tau.

---

other diseases; in Pick's disease (PiD), filaments are composed of only 3R tau isoforms; and in progressive supranuclear palsy (PSP), corticobasal degeneration (CBD), globular glial tauopathy (GGT), argyrophilic grain disease (AGD), and other tauopathies, filaments are composed of only 4R tau isoforms.

Tau filaments consist of a structured core made mostly of the repeat domain, with less structured N- and C-terminal regions forming the fuzzy coat (*Goedert et al., 1988*; *Wischik et al., 1988a*). Previously, we and others used electron cryo-microscopy (cryo-EM) to determine the atomic structures of the cores of tau filaments extracted from the brains of individuals with a number of neurodegenerative diseases (*Fitzpatrick et al., 2017*; *Falcon et al., 2018a*; *Falcon et al., 2019*; *Falcon et al., 2019*; *Arakhamia et al., 2021*; *Zhang et al., 2020*; *Shi et al., 2021*). Whereas, tau filaments from different individuals with a given disease share the same structures, different diseases tend to be characterised by distinct tau folds. Identical protofilaments can arrange in different ways to give rise to distinct filaments (ultrastructural polymorphs). In AD, two protofilaments can combine in a symmetrical way to form paired helical filaments (PHFs), or in an asymmetrical manner to form straight filaments (SFs). The extent of the ordered cores of the protofilament folds explains the isoform composition of tau filaments. Alzheimer and CTE folds consist almost entirely of R3 and R4; the Pick fold comprises most of R1, as well as all of R3 and R4; and the CBD, PSP, GGT, and AGD folds comprise the whole of R2, R3, and R4. All known tau folds also contain residues 369–379/380 from the C-terminal domain. Based on these findings, we proposed a hierarchical classification of tauopathies, which complements clinical and neuropathological characterisation, and allows identification of new disease entities (*Shi et al., 2021*). Nevertheless, it remains unclear what roles distinct tau folds play in different diseases and what factors determine structural specificity.

In human diseases, full-length tau assembles into filaments intracellularly (*Goedert et al., 2017*). Proteolytic cleavage of the fuzzy coat is characteristic of extracellular ghost tangles (*Endoh et al., 1993*). Yet, full-length recombinant tau is highly soluble and in vitro assembly into filaments requires

the addition of anionic co-factors, such as sulphated glycosaminoglycans, RNA, fatty acids, and poly-glutamate (*Goedert et al., 1996*; *Kampers et al., 1996*; *Pérez et al., 1996*; *Wilson and Binder, 1997*; *Friedhoff et al., 1998*) or the use of harsh shaking conditions with small beads and sodium azide (*Chakraborty et al., 2021*). Whether the structures of filaments assembled in vitro resemble those observed in disease requires verification by structure determination using solid-state NMR or cryo-EM. We previously showed that the addition of heparin to full-length 3R or 4R recombinant tau expressed in *Escherichia coli* led to the formation of polymorphic filaments, with structures that are unlike those in disease (*Zhang et al., 2019*). Below, we showed that the addition of RNA or phos-phoserine to full-length recombinant 4R tau also led to filament structures that are different from those observed in disease thus far.

The mechanisms of templated seeding, where filaments provide a template for the incorporation of tau monomers, thus resulting in filament growth, underlies the hypothesis of prion-like spreading of tau pathology. It is therefore possible that in vitro assembly of disease-relevant structures may also require the presence of a seed. Protein misfolding cyclic amplification (PMCA) and real-time quaking-induced conversion (RT-QuIC) are commonly used (*Soto et al., 2002*; *Saijo et al., 2020*). These techniques use protein aggregates from brain as a template to seed the assembly of filaments from recombinant proteins. However, they may not always amplify the predominant filamentous assem-blies. Therefore, structural verification of seeds and seeded aggregates is required. For α-synuclein and immunoglobulin light-chain, cryo-EM has shown that using similar amplification methods, the structures of seeded filaments differed from those of the seeds under the conditions used (*Burger et al., 2021*; *Lövestam et al., 2021*; *Radamaker et al., 2021*).

Full-length human tau is highly soluble, but truncated proteins encompassing the repeat region readily assemble into filaments that resemble AD PHFs by negative staining. A fragment consisting of residues 250–378, excluding R2, formed filaments when using the hanging drop approach, a method commonly used in protein crystallography (*Crowther et al., 1992*). The same was true of K11 and K12 constructs (tau residues 244–394 for K11, and the same sequences, excluding R2, for K12) (*Wille et al., 1992*). Moreover, proteins comprising the ordered cores of tau filaments from AD [residues 306–378], PiD [residues 254–378, but excluding R2], and CBD [residues 274–380] have been reported to assemble into filaments (*Carlomagno et al., 2021*). It remains to be established if the structures of these filaments resembled those from diseased brain. Multiple studies have also used constructs K18 and K19, which comprise the microtubule-binding repeats and four amino acids after the repeats (resi-dues 244–372 for K18, and the same sequences, excluding R2, for K19); they spontaneously assemble into filaments (*Gustke et al., 1994*; *Mukrasch et al., 2005*; *von Bergen et al., 2006*; *Li et al., 2009*; *Yu et al., 2012*; *Shammas, 2015*) and are seeding-active by RT-QuIC (*Saijo et al., 2020*). However, since all known tau structures from human brain include residues beyond 372, tau filament structures of K18 and K19 cannot be the same as those in disease. Another fragment that has been used for in vitro assembly studies is dGAE, which comprises residues 297–391 of 4R tau, and was identified as the proteolytically stable core of PHFs from tangle fragments of AD (*Wischik et al., 1988b*). It assem-bles into filaments with similar morphologies to AD PHFs by negative staining EM and atomic force microscopy (*Novak et al., 1993*; *Al-Hilaly et al., 2017*; *Al-Hilaly et al., 2020*; *Lutter et al., 2022*).

Here, we report conditions that lead to the formation of AD PHFs from purified tau (297–391) expressed in *E. coli*. We show that the same construct can also be used to form type II filaments of CTE, and demonstrate how different cations affect the differences between the structures of the two diseases. In addition, we describe to what extent the tau (297–391) fragment can be extended, as well as shortened, while still forming PHFs. We report 76 cryo-EM structures, including those of 27 previously unobserved filaments. *Table 1* gives an overview of assembly conditions (numbered) and filament types (indicated with letters). *Figure 1* and *Figure 1—figure supplements 1–7* describe the cryo-EM structures that were determined. Our results illustrate how high-throughput cryo-EM structure determination can guide the quest for understanding the molecular mechanisms of amyloid filament formation.

**Table 1.** In vitro assembly conditions for all filament types.

| Filament types | Construct (residues) | Buffer | Shaking (rpm) | Time (hr) | Fold |
|---|---|---|---|---|---|
| 1a | 297–391 | 10 mM PB 10 mM DTT pH 7.4 | 700 | 48 | New |
| 2a–d | 297–391 | 10 mM PB 10 mM DTT pH 7.4 | 200 | 48 | AD |
| 3a | 297–391 | 10 mM PB 10 mM DTT pH 7.4 0.1 µg /ml dextran sulphate | 200 | 48 | AD |
| 4a | 297–391 | 10 mM PB 10 mM DTT pH 7.4 200 mM MgCl$_2$ | 200 | 48 | AD |
| 5a | 297–391 | 10 mM PB 10 mM DTT pH 7.4 20 mM CaCl$_2$ | 200 | 48 | AD |
| 6a–c | 266/297–391* | 10 mM PB 10 mM DTT pH 7.4 | 200 | 48 | AD |
| 7a–b | 266–273 –391 | 10 mM PB 10 mM DTT pH 7.4 200 mM MgCl$_2$ | 200 | 48 | AD |
| 8a–b | 266/297–391 | 10 mM PB pH 7.4 10 mM DTT 200 mM NaCl | 200 | 48 | CTE |
| 9a–b | 266/297–391 | 10 mM PB pH 7.4 10 mM DTT 200 mM LiCl | 200 | 48 | New |
| 10a–b | 266/297–391 | 10 mM PB pH 7.4 10 mM DTT 200 mM KCl | 200 | 48 | New |
| 11a | 266/297–391 | 10 mM PB pH 7.4 10 mM DTT 100 µM ZnCl$_2$ | 200 | 48 | New |
| 12a | 266/297–391 | 10 mM PB pH 7.4 10 mM DTT 200 µM CuCl$_2$ | 200 | 48 | New |
| 13a | 266/297–391 | 10 mM PB pH 7.4 10 mM DTT 20 mM MgCl$_2$ 50 mM KCl 50 mM NaCl | 200 | 48 | New |
| 14a–b | 266/297–391 | 10 mM PB pH 7.4 10 mM DTT 20 mM MgCl$_2$ 100 mM NaCl | 200 | 48 | New |
| 15a–d | 266/297–391 | 10 mM PB pH 7.4 10 mM DTT 10 mM MgSO$_4$ 100 mM NaCl | 200 | 48 | CTE |
| 16a–b | 266/297–391 | 10 mM PB pH 7.4 10 mM DTT 10 mM NaHCO$_3$ 100 mM NaCl | 200 | 48 | New |
| 17a–c | 266/297–391 | 10 mM PB pH 7.4 10 mM DTT 500 mM NaCl | 200 | 48 | New |
| 18a | 244–391 | 50 mM PB pH 7.4 10 mM DTT 20 mM MgCl$_2$ | 200 | 76 | New |
| 19a | 244–391 | 50 mM PB pH 7.4 10 mM DTT 200 mM NaCl | 200 | 76 | New |
| 20a | 244–391 | 10 mM PB 10 mM DTT 5 mM Na$_4$P$_2$O$_7$ | 200 | 76 | New |
| 21a–b | 258–391 | 10 mM PB pH 7.4 10 mM DTT 200 mM MgCl$_2$ | 200 | 48 | New |
| 22a | 266–391 | 10 mM PB pH 7.4 10 mM DTT 200 mM MgCl$_2$ | 200 | 48 | New |
| 23a–c | 266–391 | PBS pH 7.4 10 mM DTT | 200 | 48 | CTE |
| 24a | 287–391 | 10 mM PB pH 7.4 10 mM DTT 200 mM MgCl$_2$ | 200 | 48 | AD |
| 25a | 300–391 | 10 mM PB pH 7.4 10 mM DTT 200 mM MgCl$_2$ | 200 | 48 | AD |
| 26a | 303–391 | 10 mM PB pH 7.4 10 mM DTT 200 mM MgCl$_2$ | 200 | 48 | AD |
| 27a | 305–379 | 10 mM PB pH 7.4 10 mM DTT 200 mM MgCl$_2$ | 200 | 48 | New |
| 28a | 297–421 | 10 mM PB pH 7.4 10 mM DTT 200 mM MgCl$_2$ | 200 | 48 | New |
| 29a | 297–412 | 10 mM PB pH 7.4 10 mM DTT 200 mM MgCl$_2$ | 200 | 48 | New |
| 30a | 297–402 | 10 mM PB pH 7.4 10 mM DTT 200 mM MgCl$_2$ | 200 | 48 | New |
| 31a | 297–396 | 10 mM PB pH 7.4 10 mM DTT 200 mM MgCl$_2$ | 200 | 48 | New |
| 32a–b | 297–394 | 10 mM PB pH 7.4 10 mM DTT 200 mM MgCl$_2$ | 200 | 48 | AD |
| 33a | 297–384 | 10 mM PB pH 7.4 10 mM DTT 200 mM MgCl$_2$ | 200 | 48 | AD |
| 34a–b | 297–394 | 10 mM PB pH 7.4 10 mM DTT | 700 | 48 | AD |
| 35a–d | 297–394 | PBS pH 7.4 10 mM DTT | 700 | 48 | New |
| 36a–c | 300–391 | PBS pH 7.4 10 mM DTT | 700 | 48 | New |
| 37a | 303–391 | PBS pH 7.4 10 mM DTT | 700 | 48 | New |
| 38a | 258–391 | 10 mM PB pH 7.4 10 mM DTT | 700 | 48 | GGT |
| 39a–b | 258–391 | 10 mM PB pH 7.4 10 mM DTT 5 mM phosphoglycerate | 700 | 48 | GGT |
| 40a | 258–391 | 10 mM PB pH 7.4 10 mM DTT 300 ug/ul heparan sulphate | 700 | 48 | GGT |

*Table 1 continued on next page*

*Table 1 continued*

| Filament types | Construct (residues) | Buffer | Shaking (rpm) | Time (hr) | Fold |
|---|---|---|---|---|---|
| 41a | 258–391 | 10 mM PB pH 7.4 10 mM DTT 0.1% NaN$_3$ | 700 | 48 | New |
| 42a–b | 297–408 4-pmm* | 10 mM PB pH 7.4 10 mM DTT 200 mM MgCl$_2$ | 200 | 48 | AD |
| 43a | 297–441 4-pmm | 10 mM PB pH 7.4 10 mM DTT 200 mM MgCl$_2$ | 200 | 48 | AD |
| 44a | 266–391 S356D | 10 mM PB pH 7.4 10 mM DTT 200 mM KCl | 200 | 48 | New |
| 45a | 266–391 S356D | 10 mM PB pH 7.4 10 mM DTT 200 mM NaCl | 200 | 48 | New |
| 46a | 0N4R | PBS pH 7.4 5 mM TCEP 50 ug/mL polyA RNA | 200 | 96 | New |
| 47a | 0N4R | PBS pH 7.4 5 mM TCEP 5 mM L-phosphoserine | 200 | 96 | New |

DTT: 1,4-dithiothreitol **PB:** Na$_2$HPO$_4$, NaH$_2$PO$_4$; **Dextran sulphate:** molecular weight 7–20 kDa (9011-18-1, Sigma-Aldrich). **Heparan sulphate:** 50–200 disaccharide units (57459-72-0, Sigma-Aldrich). Poly-A RNA (26763-19-9, Sigma-Aldrich); **PBS:** Phospho-buffered saline; **TCEP:** Tris(2-carboxyethyl) phosphine; *4-pmm:* four-phospho mimetic mutations: S396D S400D T403D S404D *PB: Na$_2$HPO$_4$, NaH$_2$PO$_4$,*266/297–391:* 50:50 ratio of 266LKHQ269 (3 R) and 297IKHV300 (4 R) –391. Fold: **AD:** Alzheimer's Disease protofilament fold, **CTE:** chronic traumatic encephalopathy protofilament fold and **GGT:** globular glial tauopathy-like fold, **New:** new tau fold.

## Results

### In vitro assembly of tau (297–391) into PHFs

We first performed in vitro assembly of tau (297–391) in 10 mM phosphate buffer (PB) containing 10 mM dithiothreitol (DTT), with shaking at 700 rpm, as described (*Al-Hilaly et al., 2017*). We will refer to this as assembly condition 1. Filaments formed within 4 hr, as indicated by ThT fluorescence. Cryo-EM imaging after 48 hr revealed a single type of filament comprising two protofilaments that were related by pseudo-2$_1$ helical screw symmetry. Although the extent and topology of the ordered cores resembled those of the protofilaments of AD and CTE, the protofilament cores were extended, rather than C-shaped (*Figure 2A–C*; filament type 1a).

We then reduced the shaking speed to 200 rpm. This resulted in a slower assembly reaction, with filaments appearing after 6 hr. Cryo-EM structure determination after 48 hr showed that the filaments were polymorphic. They shared the AD protofilament fold, spanning residues 305–378. However, besides AD PHFs, we also observed filaments comprising three or four protofilaments, which we called triple helical filaments (THFs) and quadruple helical filaments (QHFs) (*Figure 2D*; filament types 2b–d). THFs and QHFs have not been observed in brain extracts from individuals with AD (*Fitzpatrick et al., 2017*; *Falcon et al., 2018b*), possibly because the presence of the fuzzy coat would hinder their formation. SFs were not seen. THFs consist of a PHF and an additional single protofilament with the AD fold, whereas QHFs are made of two stacked PHFs that come in two different arrangements (types 1 and 2). The cryo-EM maps of QHF type 1 filaments were of sufficient quality to build atomic models. QHF type 1 and the THF share a common interface, where one protofilament forms a salt bridge at E342 with K343 from the third, adjoining protofilament. This protofilament remains on its own in THFs, whereas it forms a typical PHF interface with a fourth protofilament in QHFs. Although the cryo-EM reconstruction of the QHF type 2 filament was of insufficient resolution for atomic modelling, the cross-section perpendicular to the helical axis suggested that a salt bridge was present between E342 from one PHF and K321 from another (*Figure 1—figure supplement 1*; filament types 2b–d).

Next, we sought to optimise the assembly conditions. Since THFs and QHFs have not been observed in AD, their formation would confound the use of this assembly assay for screens or to model diseases. In addition, filaments from the above experiments tended to stick together, which complicated their cryo-EM imaging, and could interfere with their use. Over longer incubation periods (>76 hr), filaments tended to precipitate, resulting in cloudy solutions. To assemble tau (297–391) into pure PHFs and reduce stickiness, we explored the addition of salts and crowding reagents. In particular, the addition of 200 mM MgCl$_2$, 20 mM CaCl$_2$, or 0.1 µg/ml dextran sulphate resulted in purer populations of PHFs (~95%) (*Figure 2E*; filament types 3–5). Cryo-EM structure determination of filaments made using these conditions confirmed that their ordered cores were identical, with a root mean square deviation (r.m.s.d) of all non-hydrogen atoms of 1.3 Å, to those of AD PHFs (*Figure 2F–G*).

Tau (297–391) comprises the C-terminal 9 residues of R2, the whole of R3 and R4, as well as 23 amino acids after R4. The equivalent 3R tau construct, which lacks R2, begins with the C-terminal 9 residues of R1, the first four of which ($^{266}$LKHQ$^{269}$) are different from R2. We also assembled tau

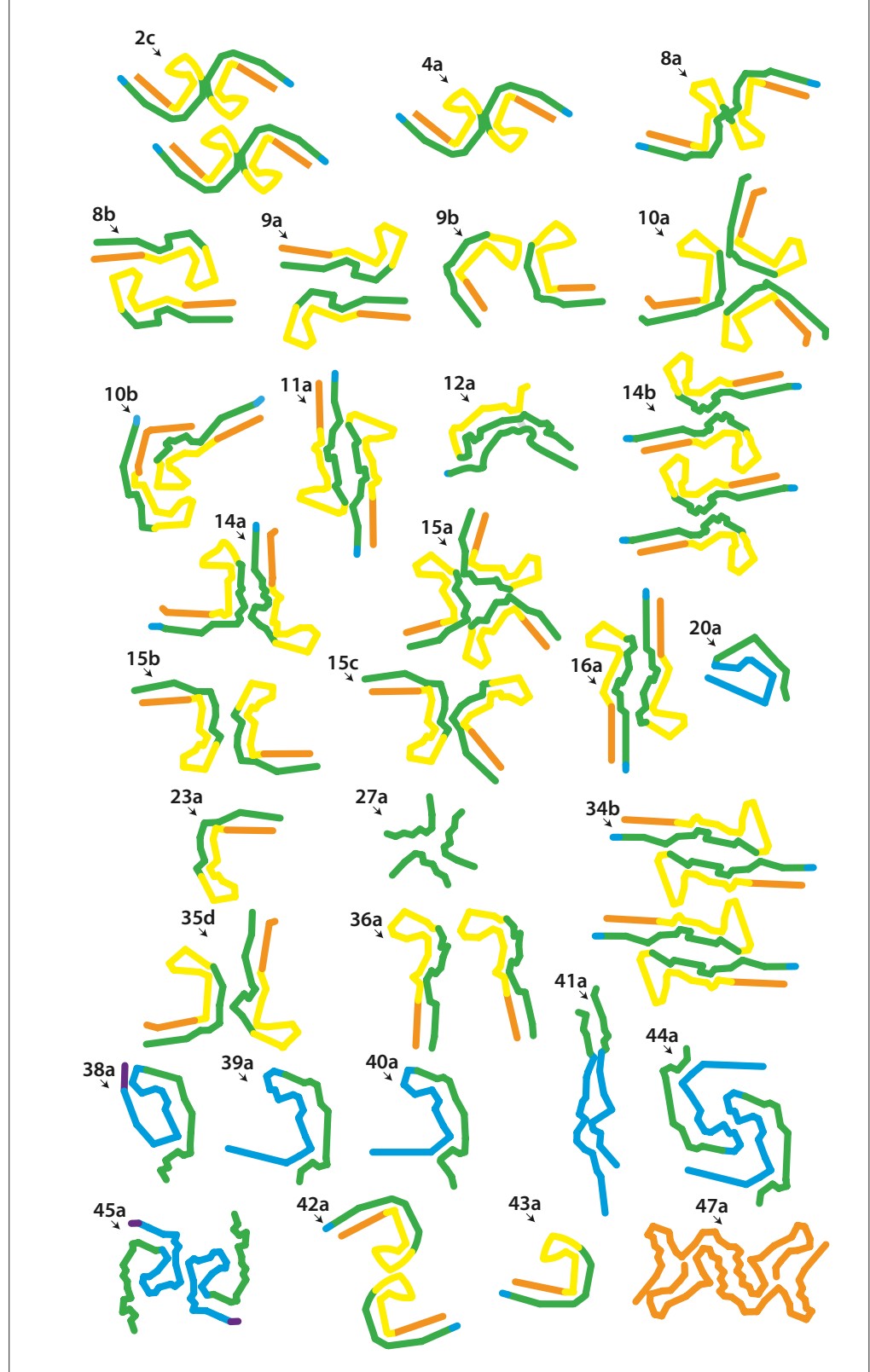

**Figure 1.** New electron cryo-microscopy (cryo-EM) structures. Backbone traces for filaments with previously unobserved structures. Residues 244–274 (R1) are shown in purple; residues 275–305 (R2) are shown in blue; residues 306–336 (R3) are shown in green; residues 337–368 (R4) are shown in yellow; residues 369–441 (C-terminal domain) are shown in orange. The filament types (as defined in **Table 1**) are shown at the top left of each structure.

*Figure 1 continued on next page*

*Figure 1 continued*

The online version of this article includes the following figure supplement(s) for figure 1:

**Figure supplement 1.** Cross-sections of electron cryo-microscopy (cryo-EM) reconstructions.

**Figure supplement 2.** Electron cryo-microscopy (cryo-EM) maps and models of new structures (part 1).

**Figure supplement 3.** Electron cryo-microscopy (cryo-EM) maps and models of new structures (part 2).

**Figure supplement 4.** Schematics of the tau folds (part 1).

**Figure supplement 5.** Schematics of the tau folds (part 2).

**Figure supplement 6.** Fourier shell correlation curves (part 1).

**Figure supplement 7.** Fourier shell correlation curves (part 2).

**Figure supplement 8.** Handedness of structure 41a.

---

(266–391), excluding R2, in the presence of 200 mM $MgCl_2$, as well as a 50:50 mixture of the 3R/4R tau constructs, in the absence of $MgCl_2$. We observed PHFs, THFs, and QHFs in the absence of $MgCl_2$, whereas assembly in the presence of 200 mM $MgCl_2$ gave rise to AD PHFs with a purity greater than 94% (*Figure 1—figure supplement 1*; filament types 6 and 7). These findings show that *bona fide* PHFs can be formed from only 3R or 4R, albeit truncated, tau. In AD and CTE, all six isoforms, each full-length, are present in tau filaments (*Goedert et al., 1992*; *Schmidt et al., 2001*). It remains to be determined if there are PHFs in human diseases that are made of only 3R or 4R tau.

The cryo-EM structures of in vitro assembled PHFs and of AD PHFs shared the same left-handed twist and pseudo-$2_1$ helical screw symmetry. Moreover, there were similar additional densities in front of lysine residues 317 and 321, and on the inside of the protofilament's C, as we previously observed for AD PHFs (*Figure 2—figure supplement 1*). The assembly buffers contained only $Na_2HPO_4$, $NaH_2PO_4$, $MgCl_2$, and DTT. Although we cannot exclude the possibility that negatively charged co-factors may have purified together with recombinant tau, it appears more likely that the additional densities arose from phosphate ions in the buffer. The phosphates' negative charges may have counteracted the positive charges of stacked lysines. It remains to be established if similar densities in AD PHFs also correspond to phosphate ions, or if other negatively charged co-factors or parts of the fuzzy coat may play a role. The fuzzy coat, which consists of only a few residues on either side of the ordered core, is not visible in cryo-EM micrographs of in vitro assembled tau (297–391) (*Figure 2—figure supplement 1*).

## The effects of salts on tau filament assembly

During optimisation of assembly, we noticed that different cations in the buffer caused the formation of filaments with distinct protofilament folds. Besides $MgCl_2$ and $CaCl_2$, which led to the formation of AD PHFs, we also explored the effects of $ZnCl_2$, $CuCl_2$, NaCl, LiCl, and KCl (*Figure 3a*). Addition of $ZnCl_2$ resulted in the same fold as observed for filaments assembled using condition 1, whereas addition of $CuCl_2$ led to folds with little resemblance to previously observed tau folds (*Figure 1* and *Figure 1—figure supplement 1*; filament types 11 and 12). $Cu^{2+}$ ions led to the formation of intermolecular disulphide bonds that were part of the ordered cores of these filaments.

Monovalent cations modulated the formation of protofilament folds that were similar or identical to AD and CTE folds. The CTE fold is similar to the AD fold, in that it also comprises a double-layered arrangement of residues 274/305–379; however, it adopts a more open C-shaped conformation and comprises a larger cavity at the tip of the C (residues 338–354), which is filled with an additional, unknown density (*Falcon et al., 2019*). We first describe how different monovalent cations led to the formation of both C-shaped and more extended protofilament folds. We then present the effects of cations on the additional density in the cavity and the conformations of the surrounding residues.

Addition of 200 mM NaCl led to the formation of two types of filaments. The first type was identical, with an all-atom r.m.s.d. of 1.4 Å to CTE type II filaments (*Figure 3A–C*; filament type 8a); in the second type (filament type 8b), two identical protofilaments with a previously unobserved, extended protofilament fold packed against each other with pseudo-$2_1$ helical symmetry. This fold resembled the extended fold observed when using condition 1. The extended fold concurred with a flipping of the side chains of residues 322–330, which were alternatively buried in the core or solvent-exposed in opposite manner to the CTE fold. Side chains before and after $^{364}$PGGG$^{367}$ had the same orientations, but formed a 90° turn in the CTE fold and adopted a straight conformation in the extended fold.

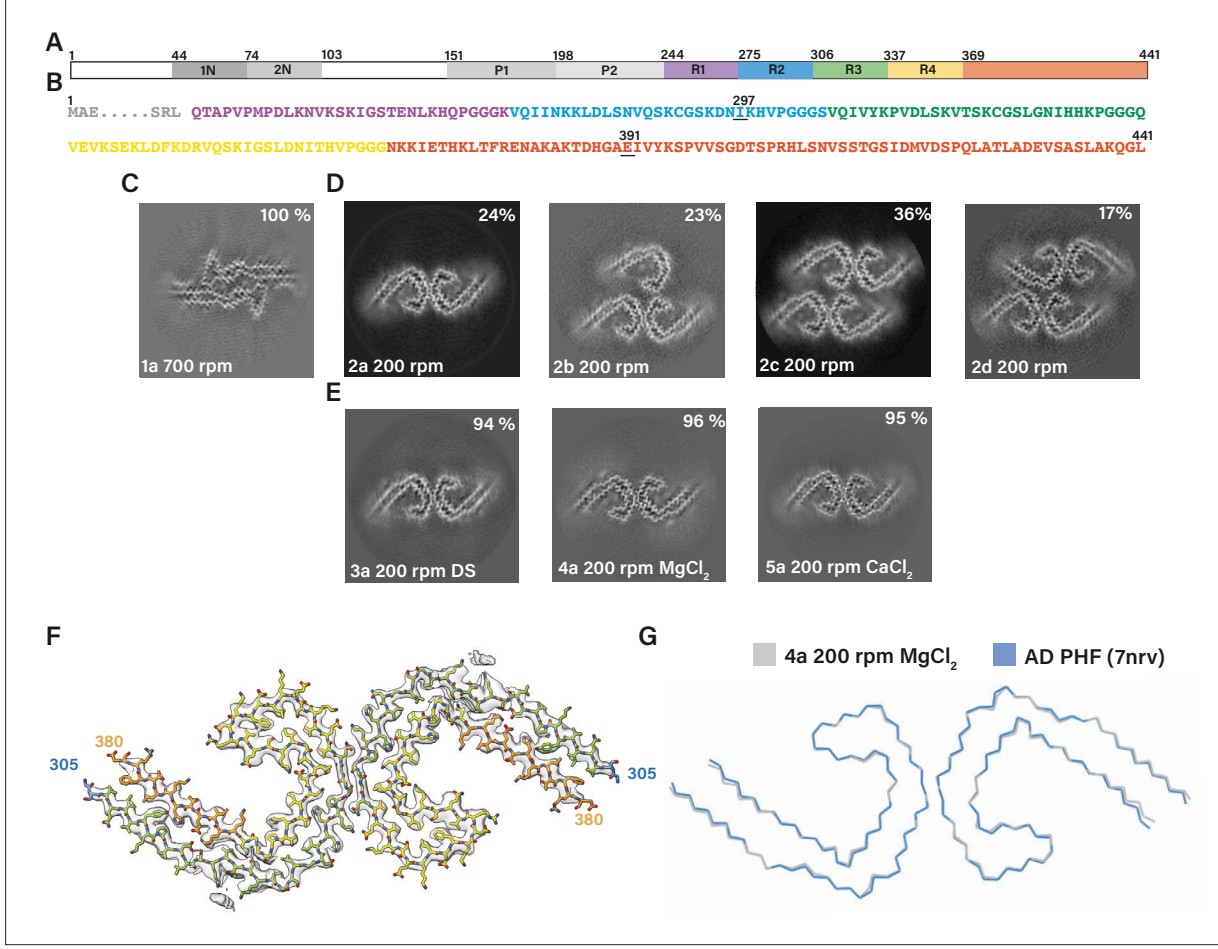

**Figure 2.** Assembly of recombinant tau into filaments like Alzheimer's disease paired helical filaments (AD PHFs). (**A**) Schematic of 2N4R tau sequence with domains highlighted. The regions 1N (44–73), 2N (74–102), P1 (151–197), and P2 (198–243) are shown in increasingly lighter greys; R1 (244–274) is shown in purple; R2 (275–305) is shown in blue; R3 (306–336) is shown in green; R4 (337–368) is shown in yellow; the C-terminal domain (369–441) is shown in orange. (**B**) Amino acid sequence of residues 244–441 of tau, with the same colour scheme as in A. (**C–E:**) Projected slices, with a thickness of approximately 4.7 Å, orthogonal to the helical axis for several cryo-microscopy (cryo-EM) reconstructions. The filament types (as defined in *Table 1*) are shown at the bottom left and the percentages of types for each cryo-EM data set are given at the top right of the images. (**C**) Conditions of *Al-Hilaly et al., 2017*, with shaking at 700 rpm. (**D**) Our adapted protocol, using 200 rpm shaking. From left to right; paired helical filament (PHF), triple helical filament, quadruple helical filament type 1, and quadruple helical filament type 2. (**E**) Our optimised conditions for in vitro assembly of relatively pure PHFs.( **F**) Cryo-EM density map (grey transparent) of in vitro assembled tau filaments of type 4a and the atomic model colour coded according as in A. (**G**) Backbone ribbon of in vitro PHF (grey) overlaid with AD PHF (blue).

The online version of this article includes the following figure supplement(s) for figure 2:

**Figure supplement 1.** Electron cryo-microscopy (cryo-EM) micrographs and density maps comparing in vitro assembled paired helical filaments (PHFs) and Alzheimer's disease paired helical filaments (AD PHFs).

Residues 338–354 had identical conformations, with an all-atom r.m.s.d for these residues of 1.3 Å, at the tips of the C-shaped and extended folds (*Figure 1*; *Figure 3—figure supplement 1*). When adding 200 mM LiCl, we observed two types of filaments, with either C-shaped or more extended protofilament folds (*Figure 3A*; filament types 10a and 10b). In the first type, two C-shaped protofilaments packed against each other in an asymmetrical manner. In the second type, two protofilaments with an extended conformation packed against each other with pseudo-$2_1$ helical symmetry. As observed for the filaments obtained with NaCl, the side chain orientations of residues 322–330 differed between folds. However, whereas the side chain of H330 was buried in the core of the C-shaped protofilament formed with NaCl, it was solvent-exposed in the C-shaped protofilament formed with LiCl. This suggests that the conformation of the $^{364}$PGGG$^{367}$ motif defines the extended or C-shaped conformation (*Figure 3—figure supplement 2*). Addition of 200 mM KCl also led to two different filaments

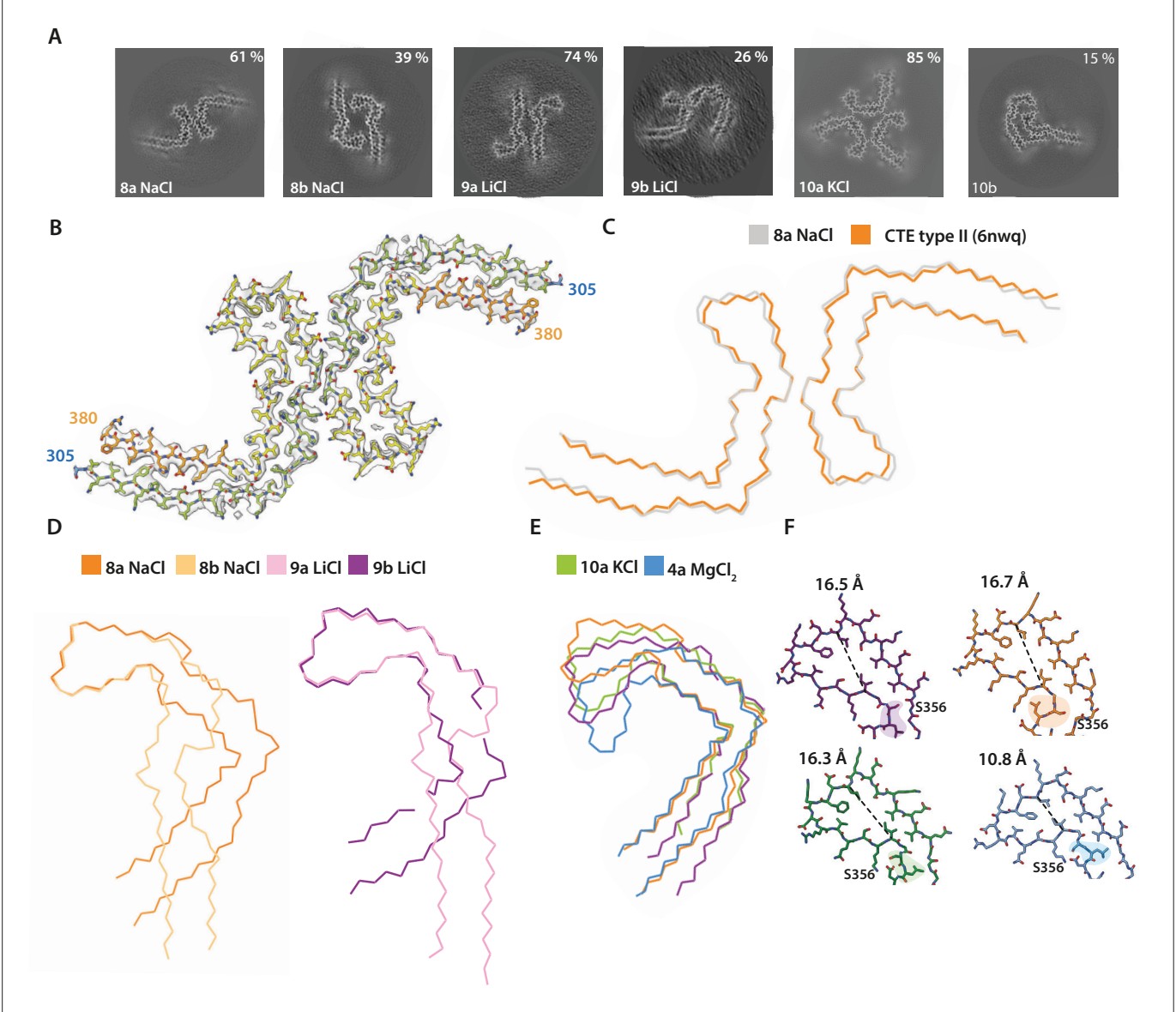

**Figure 3.** Assembly of recombinant tau into filaments like chronic traumatic encephalopathy (CTE) type II filaments. (**A**) Projected slices, with a thickness of approximately 4.7 Å, orthogonal to the helical axis are shown for different assembly conditions and filament types (as defined in *Table 1*), which are indicated in the bottom left. The percentages of types are shown in the top right of each panel. ( **B**) Cryo-EM density map (grey transparent) of filament type 8a and the corresponding atomic model with the same colour scheme as in *Figure 1*. (**C–E**) Backbone ribbon views of protofilament and filament folds. (**C**) In vitro NaCl filament type 8a (grey) overlaid with CTE type II (orange). (**D**) Extended and C-shaped protofilaments aligned at residues 338–354 for LiCl, filament types 9a and 9b (left) and NaCl, filament types 8a and 8b (right). (**E**) Filament types 8b (NaCl), 9a (LiCl), 10a (KCl), and 4a (MgCl₂) aligned at residues 356–364. (**F**) Atomic view of residues 334–358. The distance between the Cα of L344 and I354 is indicated. Filament types 8a, 8b, 9a, 9b, and 10a are shown in light purple, dark purple, dark orange, light orange, and blue, respectively.

The online version of this article includes the following figure supplement(s) for figure 3:

**Figure supplement 1.** Extended and C-shaped NaCl protofilaments.

**Figure supplement 2.** Extended and C-shaped LiCl protofilaments.

**Figure supplement 3.** Cryo-microscopy (cryo-EM) densities inside the cavities of KCl and NaCl filaments.

with either extended or C-shaped protofilament folds. However, in this case, low numbers of filaments with extended protofilaments resulted in poor cryo-EM reconstructions. The filaments with C-shaped folds comprised three protofilaments, which packed against each other with C3 symmetry (*Figure 1*; *Figure 3A*; filament type 10a).

For each monovalent cation, residues 338–354 adopted identical conformations when comparing extended and C-shaped protofilament folds (*Figure 3D*). These residues surrounded the cavity at the tip of the fold, which was filled with an additional density in the CTE fold. Additional densities were also observed in filaments formed in the presence of NaCl, KCl, and in the extended filaments formed with LiCl. The cryo-EM reconstructions of the threefold symmetric filaments formed with KCl, with a resolution of 1.9 Å, showed multiple additional spherical densities inside the protofilament core. Besides additional densities for what were probably water molecules in front of several asparagines and glutamines, the cavity at the tip of the fold contained two larger, separate spherical densities per β-rung, which were 3.1 Å apart, and at approximately 3.0–4.5 Å distance from S341 and S352, the only polar residues in the cavity. Another pair of additional densities, similar in size to those inside the cavity, was present at a distance of 2.6 Å from the carbonyl of G335 (*Figure 3—figure supplement 3*). Below, we will argue that these densities corresponded to pairs of $K^+$ and $Cl^-$ ions. Reconstructions for the filaments formed with NaCl were at resolutions of 2.8 and 3.3 Å. The additional density in these maps was not separated into two spheres, but was present as one larger blob per rung, with separation between blobs along the helical axis. Filaments formed with LiCl were resolved to resolutions of 3.1 and 3.4 Å. No additional densities were present inside the cavity of the C-shaped fold, but the cavity in the extended fold contained a spherical density that was smaller than the densities observed for NaCl and KCl filaments (*Figure 3—figure supplement 3*). Different cations also led to conformational differences in residues S356 and L357, which were akin to the differences observed previously between AD and CTE folds (*Figure 3E, F*; *Falcon et al., 2019*). In the AD fold, S356 is solvent-exposed and L357 is buried inside the protofilament core, whereas they adopt opposite orientations in the CTE fold. As mentioned, filaments formed with NaCl are identical to CTE filaments; in filaments formed with KCl, S356 and L357 are oriented in the same directions as in the AD fold; in filaments formed with LiCl, both residues are buried in the core.

Efforts to generate AD SFs using mixtures of salts were unsuccessful, but they did reveal novel protofilament interactions (*Figure 1*; *Figure 1—figure supplement 1*; filament types 13–17). Notably, using a buffer with $MgSO_4$ and NaCl, we obtained a minority of filaments with an SF interface (11%, filament type 15d). Probably because of the presence of NaCl, protofilaments adopted the CTE fold. Further exploration of the role of salts may lead to the assembly of recombinant tau into AD SFs and CTE type I filaments.

## The effects of protein length on tau filament assembly

We investigated if the N- and C-termini of tau (297–391) are required for its assembly into PHFs. We first made a series of protein fragments ending at residue 391, to explore the effects of the position of the N-terminus. Next, keeping the N-terminus at residue 297, we explored the effects of the position of the C-terminus (*Figure 4A*). Each recombinant tau fragment was assembled in 10 mM PB, 10 mM DTT, and 200 mM $MgCl_2$, at 200 rpm shaking. We assessed the presence of tau filaments by negative stain EM and used cryo-EM to determine their structures (*Figure 4B*; filament types 18–30).

Proteins comprising the entire N-terminal domain (residues 1–391 of 0N4R tau) did not assemble into filaments. The same was true of proteins starting at residues 151, 181, or 231 in the proline-rich region. When using proteins starting at 244, the first residue of R1, we observed the formation of filaments. The quality of the cryo-EM reconstructions was not sufficient for atomic modelling, but the ordered filament cores adopted a more open conformation than in the Alzheimer fold, with two protofilaments interacting at their tips. Residues 338–354 probably adopted the same conformation as in the Alzheimer fold. Addition of NaCl or pyrophosphate did not lead to the formation of PHFs or CTE filaments (*Figure 1*; *Figure 1—figure supplement 1*; filament types 19–20). However, it is possible that this construct will form filaments with the Alzheimer fold after further optimisation of assembly conditions. Proteins that started at residues 258 or 266 in R1 formed PHFs. Adding NaCl to tau (297-391), or using phosphate-buffered saline (PBS), gave rise to CTE type II filaments, as well as to filaments consisting of either single CTE protofilaments or two protofilaments, related by C2 symmetry, which formed an interface at residues 327–336 (*Figure 1—figure supplement 1*; filament types 15b, 23b). For all filaments assembled in the presence of NaCl, we saw densities inside the tip of the fold's cavity. Moreover, with the proteins starting at residues 258 and 266, we observed proteinaceous densities, which packed against the N-terminus of the protofilaments. This may have been the extension into R2 (*Figure 1—figure supplement 1*; filament types 23a–c).

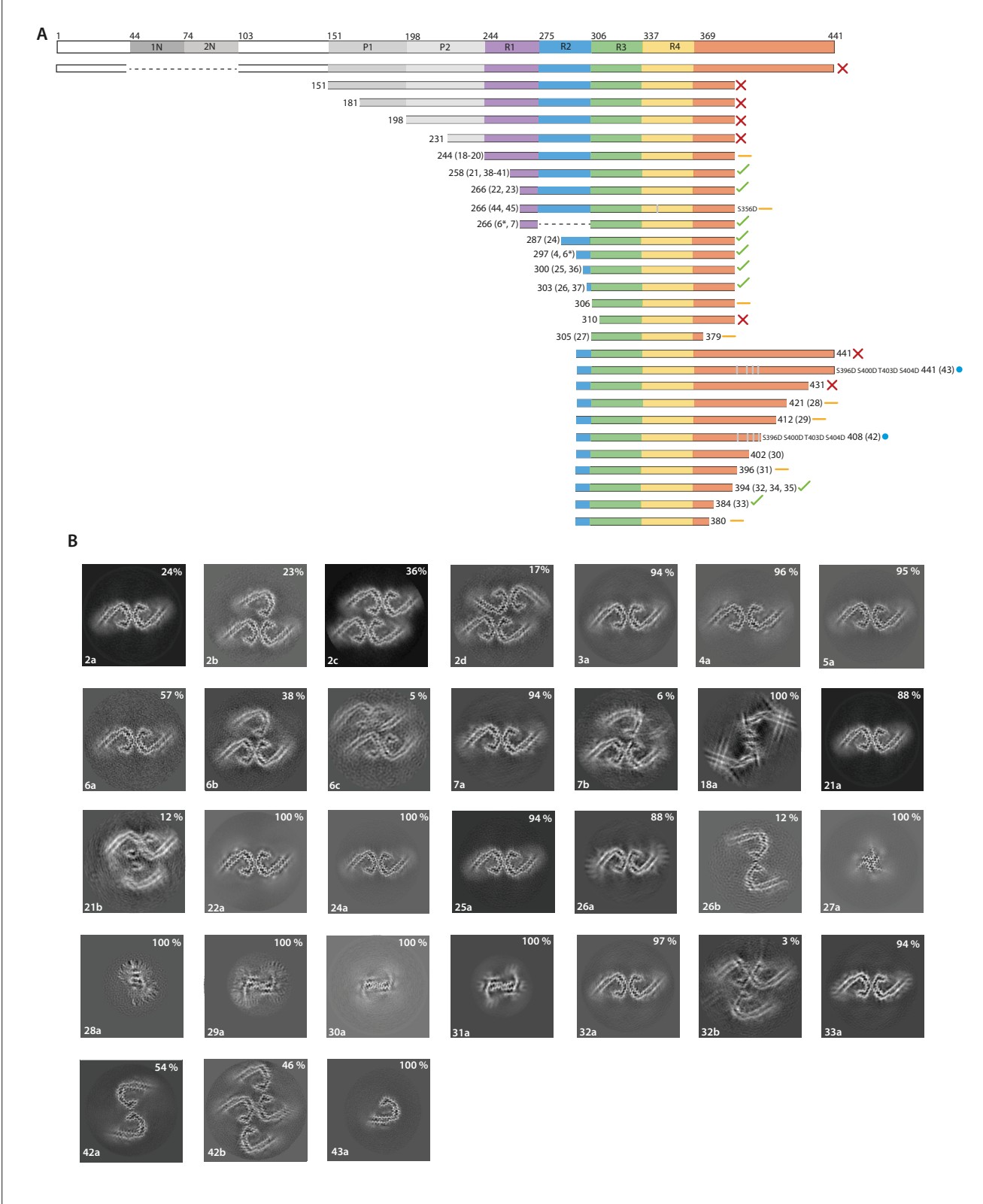

**Figure 4.** The effects of protein length on tau filament assembly. (**A**) Schematic representation of 2N4R tau and the constructs used in this study. A red cross indicates that no filaments were formed; an orange dash indicates that filaments with structures distinct from Alzheimer's disease paired helical filaments (AD PHFs) were formed; a blue circle indicates that the AD protofilament fold was formed; a green tick indicates that AD PHFs were formed. Resulting filament types (as defined in *Table 1*) are indicated for each experiment. (**B**) Projected slices, with a thickness of approximately 4.7 Å,

*Figure 4 continued on next page*

*Figure 4 continued*

orthogonal to the helical axis for the filaments formed with the constructs in A. Filament types are indicated at the bottom left; percentages of filament types in each cryo-EM data set are shown in the top right. (*) 50:50 ratio of 266–391 (3R) and 297–391 (4R) tau.

The online version of this article includes the following figure supplement(s) for figure 4:

**Figure supplement 1.** Comparison of in vitro assembled tau filaments with the globular glial tauopathy (GGT) fold.

**Figure supplement 2.** Coomassie stained sodium dodecyl-sulfate polyacrylamide gel electrophoresis (SDS-PAGE) gel for constructs 297–391, 297–408, and 296–441 before and after filament assembly.

Tau (297–391) can also be shortened from the N-terminus, as proteins starting at residues 300 or 303 still formed filaments with AD or CTE folds (*Figure 1—figure supplement 1*; filament types 25–26). When shortened to residue 306, we also observed filaments, but their stickiness precluded cryo-EM structure determination. No filaments were formed when shortening the protein to residue 310. It has previously been shown that the PHF6 motif ($^{306}$VQIVYK$^{311}$ in tau) is essential for filament formation (*von Bergen et al., 2000*; *Li and Lee, 2006*).

Reminiscent of what we observed for the N-terminal domain, the tau starting at residue 297 and ending at 441 failed to assemble into filaments. The same was true of tau (297–431). However, proteins ending at residues 421, 412, 402, or 396 formed filaments, but their ordered cores are much smaller than in the Alzheimer fold, precluding cryo-EM structure determination to sufficient resolution for unambiguous atomic modelling (*Figure 4B*; filament types 28–31). We did observe filaments with the Alzheimer fold for a protein ending at residue 394 (*Figure 1—figure supplement 1*; *Figure 4B*; filament type 32).

As observed for proteins ending at residue 384, when tau (297-391) was shortened from the C-terminus, it could still form AD and CTE folds (*Figure 4B*; filament type 33). However, constructs ending at residue 380 formed straight ribbons, precluding cryo-EM structure determination.

As shown in *Figure 2*, we found that reducing the shaking speed from 700 to 200 rpm was important for the formation of filaments with the AD fold by tau (297–391). To show that this was not only the case for tau (297–391), we assembled tau (258–391) and tau (297–394) with a shaking speed of 700 rpm. For tau (297–394), besides structures similar to those formed by tau (297–391) at 700 rpm, we observed an additional filament with pseudo-$2_1$ helical screw symmetry that comprised four protofilaments (*Figure 1*; *Figure 1—figure supplement 1*; filament type 34). In the presence of NaCl, we observed filaments with similar extended conformations, but with a CTE cavity. This was also the case when tau (300–391) and tau (303–391) were assembled at 700 rpm in PBS (*Figure 1*; *Figure 1—figure supplement 1*; filament types 35–37). When assembling tau (258–391) at 700 rpm, we observed protofilaments with partial similarity to the GGT fold. The structures were nearly identical at residues 288–322, with an all-atom r.m.s.d. of 1.5 Å, (*Figure 1*; *Figure 1—figure supplement 1*; *Figure 4—figure supplement 1*; filament type 38a). It is possible that the absence of a non-proteinaceous co-factor, which was hypothesised to be present inside the GGT fold, precluded formation of *bona fide* GGT filaments. Addition of heparan sulphate or phosphoglycerate to the assembly buffer did not result in formation of the GGT fold (*Figure 1*; *Figure 4—figure supplement 1*; filament types 39–41). Further experiments are needed to identify the co-factor of GGT.

## The effects of pseudophosphorylation on tau filament assembly

Whereas in vitro filament assembly of recombinant tau repeats was inhibited by N- and C-terminal regions, it is predominantly or only full-length tau that assembles into filaments in AD (*Lee et al., 1991*; *Goedert et al., 1992*). This difference may be due to the fact that we used unmodified recombinant proteins for in vitro assembly, while tau undergoes extensive post-translational modifications in AD (*Wesseling et al., 2020*). Immunochemistry and mass spectrometry have identified abnormal hyperphosphorylation of S396, S400, T403, and S404 in PHF-tau (*Lee et al., 1991*; *Hasegawa et al., 1992*; *Kanemaru et al., 1992*; *Morishima-Kawashima et al., 1995*). Because these sites are located in the C-terminal domain that prevents the formation of PHFs from recombinant tau, we hypothesised that their phosphorylation may modulate PHF formation and help to overcome the inhibitory effects of the C-terminal domain. We therefore mutated these residues to aspartate, with its negative charge mimicking the negative charges of phosphorylation, in the tau constructs ending at residues 408 and 441 (*Figure 4*; filament types 42–43).

As described above, tau (297–408) formed filaments with small ordered cores; its pseudophosphorylated mutant formed two types of filaments consisting of AD PHF protofilaments. Both filament types had the same inter-protofilament interface as that of THFs and type 1 QHFs, where E342 from one protofilament forms a salt bridge with K343 of the other. The second filament type was a doublet of the first. Importantly, tau (297–441), which does not assemble into filaments, did so in presence of the four phospho-mimetic mutations. Filaments consisted of a single protofilament with the AD fold.

It has been suggested that AD filaments may also be phosphorylated at S356 (*Hanger et al., 1998*; *Hanger et al., 2007*). We therefore tested phospho-mimetic mutation S356D. However, using tau (266–391) with S356D, we were unable to form PHFs. Instead, filaments consisted of two asymmetric protofilaments, both of which had features reminiscent of the GGT fold. In the presence of NaCl, this construct formed filaments with a novel fold, consisting of two identical protofilaments related by pseudo-$2_1$ helical symmetry (*Figure 1*; *Figure 1—figure supplement 1*; *Figure 4—figure supplement 1*; filament types 44 and 45). Analysis by gel electrophoresis confirmed that the tau constructs forming these filaments remained intact (*Figure 4—figure supplement 2*).

## The effects of anionic co-factors on tau filament assembly

Addition of anionic co-factors leads to the formation of filaments from full-length tau (*Goedert et al., 1996*; *Kampers et al., 1996*; *Kampers et al., 1996*; *Pérez et al., 1996*; *Wilson and Binder, 1997*; *Farid et al., 2014*; *Chakraborty et al., 2021*). However, we previously showed that heparin-induced tau filament formation led to structures that were different from those observed in disease thus far, under the conditions used (*Zhang et al., 2019*). To further explore the effects of anionic co-factors, we solved cryo-EM structures of filaments formed using full-length recombinant tau in the presence of RNA (*Kampers et al., 1996*) or phosphoserine. The resulting tau filaments had structures that have not been observed previously.

Addition of RNA led to the formation of two asymmetrical and extended protofilaments, similar to heparin-induced 3R tau filaments (*Zhang et al., 2019*; *Figure 1—figure supplement 1*; filament type 46). The resulting map was of insufficient resolution for atomic modelling. We calculated a 1.8 Å resolution reconstruction for full-length tau filaments that formed in the presence of 5 mM phosphoserine (*Figure 1—figure supplement 1*; filament type 47). At this resolution, the absolute handedness of the filament was obvious from the position of the main-chain carbonyl oxygens. Whereas most tau filaments described in this paper have a left-handed twist (see Methods), phosphoserine-induced filaments are right-handed. Surprisingly, the ordered core of this filament comprised residues 375–441, encompassing only residues from the C-terminal domain. Filaments consist of two protofilaments that are related by pseudo-$2_1$ helical symmetry and the protofilament fold contains eight β-strands. Several water molecules were observed, particularly in front of the side chains of serines, threonines, and asparagines. This is the first example of a region outside the microtubule-binding repeats of tau forming amyloid filaments.

## Discussion

In the course of protein evolution, natural selection has produced amino acid sequences that fold into specific protein structures to fulfil the multitude of tasks required to maintain life. The observation that same proteins can adopt multiple different amyloid structures has made it clear that the paradigm by which a protein's sequence defines its structure may not hold for amyloids. Apparently, the packing of β-sheets against each other in amyloid filaments can happen in many ways for a single amino acid sequence. Our work described here and performed previously (*Zhang et al., 2019*; *Shi et al., 2021*) illustrates this for tau. Similar observations have been made for amyloid-β (*Bertini et al., 2011*; *Lu et al., 2013*; *Xiao et al., 2015*; *Wälti et al., 2016*; *Kollmer et al., 2019*; *Yang et al., 2022*), α-synuclein (*Tuttle et al., 2016*; *Li et al., 2018a*; *Li et al., 2018b*; *Guerrero-Ferreira et al., 2019*; *Schweighauser et al., 2020*; *Lövestam et al., 2021*), TAR DNA binding protein-43 (*Arseni et al., 2022*; *Li et al., 2021*), and immunoglobulin light chain (*Radamaker et al., 2019*; *Swuec et al., 2019*). Because amyloid filaments of these proteins are typically associated with pathology, rather than function, their structural diversity could be disregarded. However, the observation that, for tau, and possibly also other proteins, the different amyloid structures define distinct neurodegenerative conditions, raises important questions on what drives their formation.

The lack of in vitro assembly models for recombinant tau that replicates the amyloid structures observed in disease has hampered progress. Here, we identified conditions for the in vitro assembly of AD PHFs and type II CTE filaments and established the formation of these structures by cryo-EM structure determination to resolutions sufficient for atomic modelling. The latter is crucial. Biochemical methods that discriminate between the different structures in solution do not exist, and negative stain EM or atomic force microscopy does not provide sufficient resolution to unequivocally distinguish between them.

The ability to make AD PHFs and CTE type II filaments in vitro opens new avenues for studying tauopathies. In vitro assembly assays could be used to screen for compounds that inhibit filament formation. Alternatively, in vitro generated filaments may be used in high-throughput screens for small-molecule compounds or biologics that bind specifically to a single type of filament. Such amyloid structure-specific binders could be developed into ligands for positron emission tomography to differentiate between tauopathies in living patients. Moreover, specific binders could be explored for use in therapeutic approaches that aim to degrade filaments inside neurons through their coupling with the protein degradation machinery inside the cell. Ultimately, specific binders could even obviate the need for cryo-EM structure determination to confirm the formation of the correct types of filaments in new model systems for disease.

Besides their use in screens, in vitro assembly of tau filaments also provides a model system, amenable to experimental perturbation, for the study of the molecular mechanisms that underlies amyloid formation. Coupled to cryo-EM structure determination, this provides a promising model system for studying the formation of different tau folds that define distinct tauopathies. The work presented in this paper provides two examples, as outlined below.

## Ideas and speculations

A major difference between Alzheimer and CTE tau folds is the presence of a larger cavity that is filled with an additional density in the CTE fold. Based on the relatively hydrophobic nature of this cavity, we previously hypothesised that this additional density may correspond to an unknown hydrophobic co-factor that assembles with tau to form CTE filaments (*Falcon et al., 2019*). However, we now observe the in vitro assembly of CTE type II filaments in the absence of hydrophobic co-factors. Instead, whether CTE type II filaments or AD PHFs formed in vitro was determined by the presence or absence of NaCl. Moreover, using different monovalent cations changed the additional densities, as well as the conformation of residues surrounding the cavity. The 1.9 Å map for the structure formed with KCl showed two spherical blobs of additional density per β-rung that were in an arrangement that would be consistent with a pair of $K^+$ and $Cl^-$ ions; a similar pair of spherical densities was also observed in front of G335 (*Figure 3—figure supplement 3*). Therefore, it is possible that the extra density in the cavity of filaments formed with NaCl also corresponds to $Na^+$ and $Cl^-$ ions. The continuous nature of the additional density along the helical axis could arise from limited resolution of the reconstructions with NaCl, or from the ions not obeying the 4.75 Å helical rise that is imposed on the reconstruction. The observation that the filaments formed with NaCl are identical to CTE type II filaments, which show a similar extra density, suggests that, rather than being filled with hydrophobic co-factors, the cavity in CTE filaments may contain NaCl. $Na^+$ and $Cl^-$ levels in neurons are typically much lower than the 200 mM NaCl used here, but it could be that brain trauma somehow leads to increased levels of these ions in the brain regions where tau filaments first form. Additional studies will explore this further.

Our results have also identified how regions outside the repeats interfere with the in vitro assembly of AD PHFs. In particular, the presence of the N-terminal domain, including its proline-rich region, and residues beyond 421 in the C-terminal domain, inhibited the spontaneous assembly of recombinant tau. In addition, the presence of residues 396–421 led to the formation of filaments with much smaller ordered cores. The C-terminal region has been reported to inhibit anionic co-factor-induced assembly of full-length tau, with pseudophosphorylation overcoming inhibition (*Abraha et al., 2000*; *Haase et al., 2004*).

Interestingly, mutating serine or threonine residues at positions 396, 400, 403, and 404 to aspartate, to mimic phosphorylation, overcame these inhibitory effects. Although we did not yet examine the effects of pseudophosphorylation of the N-terminal domain, we note that the proline-rich region upstream of the repeats is positively charged, which may inhibit amyloid formation. The

introduction of negative charges through phosphorylation, or the removal of positive charges by, for instance, acetylation, may be necessary for filament assembly. In AD, filamentous tau is extensively hyperphosphorylated, especially in the proline-rich region and the C-terminal domain (*Grundke-Iqbal et al., 1986*; *Iqbal et al., 2016*). Hyperphosphorylation inhibits the ability of tau to interact with microtubules (*Bramblett et al., 1993*; *Yoshida and Ihara, 1993*), which may be necessary for filament formation, since the physiological function of tau and its pathological assembly requires the repeat region. Other post-translational modifications, such as acetylation and ubiquitination of positively charged residues in the microtubule-binding repeats, may also play a role (*Wesseling et al., 2020*). It is therefore possible that post-translational modifications of tau lead to a higher propensity to form amyloid filaments in disease. Thus, the enzymes causing these modifications could be therapeutic targets.

Moreover, most of the structures in *Figure 1*, all of which are of recombinant tau proteins over-expressed in *E. coli*, share features with the Alzheimer and CTE folds: a cross-β packing of residues from near the start of R3 against residues from near the start of the C-terminal domain, and a turn in R4. Other tau folds are markedly different. Unmodified tau may have a tendency to form filaments that resemble Alzheimer and CTE folds, whereas specific post-translational modifications and/or associated molecules may play a role in driving the formation of tau folds associated with other tauopathies.

## Outlook

Our work illustrates how cryo-EM structure determination can guide the development of better experimental models by showing that the filaments generated from recombinant tau are identical to those formed in disease. As a start, we identified in vitro assembly conditions for the formation of tau filaments like those of AD and CTE. Future work will explore which factors, including post-translational modifications and/or associated molecules, determine the formation of other tau filaments.

Our results do not only have implications for the in vitro assembly of tau. Similar strategies could be applied to the study of other amyloid-forming proteins, and insights from in vitro assembly studies may carry over to the development of better experimental models of disease, including in cell lines and in animals. Whereas solving a cryo-EM structure previously constituted a scientific project in itself, current throughputs allow the use of cryo-EM structure determination as a tool in projects with a wider scope. We envision that this type of high-throughput cryo-EM structure determination will play a crucial role in future investigations into the structural diversity of amyloids.

# Materials and methods

## Key resources table

| Reagent type (species) or resource | Designation | Source or reference | Identifiers | Additional information |
|---|---|---|---|---|
| Recombinant DNA reagent | Plasmid: pRK172-0N4R | PMID:2124967 | NCBI Reference Sequence: NM_005910.5 | All constructs were derived from this plasmid |
| Strain, strain background (*Escherichia coli*) | BL21(DE3) | Agilent | 200,131 | Chemically competent cells |
| Software, algorithm | RELION | PMID:30412051 | RRID:SCR_016274 | RELION 4.0 Helical reconstruction |
| Software, algorithm | Coot | PMID:20383002 | RRID:SCR_014222 | Model building |
| Software, algorithm | ISOLDE | PMID:20383002 | | Model refinement |

## Cloning

Constructs were made with in vivo assembly cloning (*García-Nafría et al., 2016*) using pRK172 0N4R human tau. Reverse and forward primers were designed to share 15–20 nucleotides of homologous region and 15–30 nucleotides for annealing to the template with melting temperatures ranging from 58°C to 67°C.

## Protein expression and purification

Expression of tau was carried out in *E. coli* BL21 (DE3) gold cells, as described (**Studier, 2005**). In brief, a single colony was inoculated into 500 ml lysogeny broth (LB) auto-induction media and grown for 8 hr at 37°C, followed by subsequent expression for 16 hr at 24°C. Cells were harvested by centrifugation (4000× *g* for 20 min at 4°C), and resuspended in washing buffer (WB: 50 mM MES at pH 6.0; 10 mM EDTA; 10 mM DTT, supplemented with 0.1 mM PMSF and cOmplete EDTA-free protease cocktail inhibitors, at 10 ml/g of pellet). Cell lysis was performed using sonication (at 40% amplitude using a Sonics VCX-750 Vibracell Ultra Sonic Processor for 7 min, 5 s on/10 s off). Lysed cells were centrifuged at 20,000× *g* for 35 min at 4°C, filtered through 0.45 µm cut-off filters and loaded onto a HiTrap SP HP 5 ml column (GE Healthcare) for cation exchange. The column was washed with 10 column vol of WB and eluted using a gradient of WB containing 0–1 M NaCl. Fractions of 3.5 ml were collected and analysed by SDS-PAGE bis-TRIS 4–12% or Tris Glycine 4–20%. Protein-containing fractions were pooled and precipitated using 0.3 g/ml ammonium sulphate and left on a rocker for 30 min at 4°C. Precipitated proteins were then centrifuged at 20,000× *g* for 35 min at 4°C, and resuspended in 2 ml of 10 mM PB, pH 7.2–7.4, with 10 mM DTT, and loaded onto a 60/10 Superdex size exclusion column. Size exclusion fractions were analysed by SDS-PAGE, protein-containing fractions pooled and concentrated to 6 mg/ml using molecular weight concentrators with a cut-off filter of 3 kDa. Purified protein samples were flash frozen in 100 µl aliquots for future use.

## Filament assembly

Purified protein samples were thawed and diluted to 4 mg/ml. Filaments were assembled in a FLUO-star Omega (BMG Labtech, Aylesbury, United Kingdom) using Corning 96 Well Black Polystyrene Microplate (Thermo Fisher Scientific) with orbital shaking at 37°C. See *Table 1* for duration and speed settings. The addition of compounds proceeded by first adding proteins, diluted with 10 mM PB and 10 mM DTT; salts and/or cofactors were added last. Each condition was run in two separate wells. In one well, 3 µM ThT was added to allow continuous monitoring of protein assembly. The other well, without ThT, was used for microscopy. The presence of filaments was also assessed using negative stain electron microscopy on a Thermo Fisher Scientific Tecnai Spirit (operating at 120 kV).

## Electron cryo-microscopy

Samples with confirmed filaments were centrifuged at 3000× *g* for 2 min to remove any large aggregates, and 3 µl aliquots were applied to glow-discharged R1.2/1.3, 300 mesh carbon Au grids that were plunge-frozen in liquid ethane using a Thermo Fisher Scientific Vitrobot Mark IV. Cryo-EM data were acquired at the MRC Laboratory of Molecular Biology (LMB) and at the Research and Development facility of Thermo Fisher Scientific in Eindhoven (TFS). All images were recorded at a dose of 30–40 electrons per Å$^2$, using EPU software (Thermo Fisher Scientific), and converted to tiff format using relion_convert_to_tiff prior to processing. See *Table 2* and *Supplementary file 1* Tables 1-25 for detailed data collection parameters.

**Table 2.** Data acquisition details for the different microscopes.

| Microscope | LMB Krios G1 | LMB Krios G2 | TFS Glacios | TFS Krios G4 |
|---|---|---|---|---|
| Magnification | 105,000 | 96,000 | 165,000 | 165,000 |
| Camera | K2*/K3 | Falcon 4 | Falcon 4 | Falcon 4 |
| Energy filter (eV) | 20 | NA | 10 | 10 |
| Voltage (kV) | 300 | 300 | 200 | 300 |
| Electron exposure (e–/Å$^2$) | 40 | 30/40 | 40 | 40 |
| Defocus range (µm) | 1.5–3 | 1.2–2.5 | 0.6–1.2 | 0.6–1.2 |
| Pixel size (Å) | 0.85/1.145* | 0.824 | 0.672 | 0.727 |

LMB: Laboratory of Molecular Biology.
*LMB Krios G1 initially had a K2 camera, which was later replaced by a K3 camera. Only condition 1 a (as defined in *Table 1*) was collected on the K2 camera. The pixel size on the K2 camera was 1.145 Å.

At LMB, images were recorded on a Krios G1 with a K2 or K3 camera (Gatan), using an energy slit of 20 eV on a Gatan energy filter, or on a Krios G2 with a Falcon four camera (Thermo Fisher Scientific), without an energy filter. At TFS, images were recorded on a Krios G4 with a CFEG, a Falcon four camera, and a Selectris X energy filter that was used with a slit width of 10 eV.

At TFS, cryo-EM grid screening and part of data acquisition were performed using EPU-Multigrid on a Glacios microscope equipped with a Selectris X energy filter (Thermo Fisher Scientific) and a Falcon four camera. Eight grids were loaded for each 48 hr EPU-Multigrid run. Each grid was loaded to the stage for session set-up, grid square selection, and ice-filter preparation. The session information was stored in EPU and associated with the grid position in the autoloader. After all the grids sessions were created and stored in queue, twofold astigmatism was corrected and the beam tilt was adjusted to the coma-free axis using automatic functions within EPU. Before starting the EPU Multigrid queue, and once for each 48 hr session, the Selectris X filter slit was cantered, and the filter tuned for isochromaticity, magnification, and chromatic aberrations, using Sherpa software (Thermo Fisher Scientific). During the fully automated EPU Multigrid runs, each grid was loaded onto the stage, grid squares were brought to eucentric height, and holes were selected with the stored ice filter settings. For each grid, 2000 images were collected with a 10 eV energy filter slit width. Images were collected in electron event registration (EER) mode using the aberration-free image shift method in EPU version 2.12. Under these conditions, a throughput of 200–250 images per hour was achieved depending on the number of holes available per grid square.

## Helical reconstruction

Movie frames were gain corrected, aligned, and dose weighted using RELION's motion correction program (*Zivanov et al., 2019*). Contrast transfer function (CTF) parameters were estimated using CTFFIND-4.1 (*Rohou and Grigorieff, 2015*). Helical reconstructions were performed in RELION-4.0 (*He and Scheres, 2017*; *Kimanius et al., 2021*). Filaments were picked manually or automatically using Topaz (*Bepler et al., 2019*). The neural network in Topaz was trained using a subset of 10,000 extracted segments from the dataset of construct 305–379 (see *Supplementary file 1*, table 13). This trained model was used to predict positions for filaments for all other datasets using a Topaz threshold value ranging from –4 to –7, combined with custom-made Python code to output start-end coordinates of helical segments, rather than individual coordinates for all segments (*Lövestam and Scheres, 2022*; *Scheres, 2022*). In total, 38 data sets were picked using Topaz. Data sets with low contrast, due to thick ice or imaging too close to focus, required manual picking. The picked particles were extracted in boxes of either 1028 or 768 pixels, and then down-scaled to 256 or 128 pixels, respectively. For all data sets, reference-free 2D classification was performed to assess different polymorphs, cross-over distances, and to select suitable segments for further processing. Initial models were generated de novo from 2D class average images using relion_helix_inimodel2d (*Scheres, 2020*). Subsequently, 3D classification was used to select particles leading to the best reconstructions, and 3D auto-refinement was used to extend resolution and to optimise the helical twist and rise parameters. For all rendered maps in *Figures 1, 2 and 4*, Bayesian polishing (*Zivanov et al., 2019*) and CTF refinement (*Zivanov et al., 2020*) were performed to further increase resolution. Final maps were sharpened using standard post-processing procedures in RELION, and reported resolutions were estimated using a threshold of 0.143 in the FSC between two independently refined half-maps (*Figure 1—figure supplements 6–7*; *Chen et al., 2013*).

Reported percentages of filament types were calculated from the number of assigned segments to 2D classes. Filament types 15c and 15d could not be distinguished from their 2D class averages. In this case, fuzzy density in the reconstruction hinted at remaining structural heterogeneity in the data. These filament types were separated by 3D classification, and the reported percentages were calculated from the number of segments in the 3D classes. We note that experimental noise in the cryo-EM images leads to stochasticity in their class assignments, and that sample preparation artefacts may affect different filament types to different extents. Therefore, the reported percentages might not reflect the exact relative amounts of filament types in the original assembly reaction.

## Model building

All atomic models were built de novo in COOT, using three rungs for each structure. Coordinate refinement was performed in ISOLDE (*Croll, 2018*; *Casañal et al., 2020*). Dihedral angles from the middle

rung, which was set as a template, were also applied to the rungs below and above. For each refined structure, separate model refinements were performed on the first half-map, after increasing the temperature to 300 K for 1 min. The resulting model was then compared to this half-map (FSCwork), as well as to the other (FSCtest), to confirm the absence of overfitting (*Figure 1—figure supplements 6–7*). Further details of data processing and model refinement and validation are given in *Supplementary file 1*, Tables 1-25.

For reconstructions with estimated resolutions beyond 2.9 Å, densities for the backbone oxygen atoms provide direct evidence for the handedness of the filament. In this manner, both AD and CTE filaments were shown to be left-handed (*Falcon et al., 2019*; *Shi and Murzin, 2021*). Structures that contained protofilaments with AD or CTE folds (or minor variations thereof, e.g. with LiCl or KCl instead of NaCl) were assumed to be also left-handed. With the exception of the reconstruction for filament type 47a, which showed backbone oxygen atoms indicative of a right-handed filament, all reconstructions of filaments with previously unobserved protofilament folds with resolutions beyond 2.9 Å (i.e. filament types 12a, 27a 38a, 40a, 44a, 45a, and 47a) showed backbone oxygen densities indicative of left-handed filaments. Only two filament types with previously unobserved protofilament folds were solved at resolutions worse than 2.9 Å: filament type 39a at 3.4 Å and filament type 41a at 3.2 Å. Filament type 39a is the same fold as filament type 40a, which at 2.3 Å resolution was shown to be left-handed. Filament 41a comprises a polyproline-like left-handed helix at the end of R2 that is similar to the polyproline-like helices observed at the end of R3 in PHFs and CTE filaments, suggesting that filament 41a is also left-handed. In addition, we modelled and refined this structure in both hands.

The model fitted in the left-handed map has a better FSC and appears to fit the density better than the model fitted in the right-handed map (*Figure 1—figure supplement 8*). The handedness of filaments with previously unobserved protofilament folds that were solved at resolutions insufficient for atomic modelling (i.e. filament types 19a, 28a, 29a, 30a, 31a, and 46a) remains unclear.

The schematics in *Figure 1—figure supplements 4–5* were made with Takanori Nakane's atoms2svg.py script, which is publicly available from: https://doi.org/10.5281/zenodo.4090924.

## Acknowledgements

We thank RA Crowther, JG Greener, and M Wilkinson for helpful discussions; T Darling and J Grimmett for help with high-performance computing; D Cats and M van Beers for help with the Glacios and Krios G4 systems at Eindhoven; and the Electron Microscopy Facility of the MRC Laboratory of Molecular Biology and the Thermo Fisher Scientific Research and Development Facility for help with cryo-EM data acquisition.

## Additional information

### Competing interests

Fujiet Adrian Koh, Bart van Knippenberg, Abhay Kotecha: is affiliated with Thermo Fisher Scientific. The author has no financial interests to declare. Sjors HW Scheres: Reviewing editor, *eLife*. The other authors declare that no competing interests exist.

### Funding

| Funder | Grant reference number | Author |
|---|---|---|
| Medical Research Council | MC_UP_A025_1013 | Sjors HW Scheres |
| Medical Research Council | MC-U105184291 | Michel Goedert |

The funders had no role in study design, data collection and interpretation, or the decision to submit the work for publication.

### Author contributions

Sofia Lövestam, Sofia Lovestam performed all experiments, except cryo-EM data acquisition at Eindhoven, Conceptualization, Data curation, Formal analysis, Investigation, Validation, Visualization, Writing – original draft, Writing – review and editing; Fujiet Adrian Koh, Resources, Writing – review

and editing; Bart van Knippenberg, Software, Writing – review and editing; Abhay Kotecha, Investigation, Resources, Software, Writing – original draft, Writing – review and editing; Alexey G Murzin, Formal analysis, Validation, Visualization, Writing – review and editing; Michel Goedert, Conceptualization, Data curation, Formal analysis, Funding acquisition, Project administration, Supervision, Writing – original draft, Writing – review and editing; Sjors HW Scheres, Conceptualization, Data curation, Formal analysis, Funding acquisition, Methodology, Project administration, Software, Supervision, Validation, Visualization, Writing – original draft, Writing – review and editing

### Author ORCIDs
Michel Goedert ⓘ http://orcid.org/0000-0002-5214-7886
Sjors HW Scheres ⓘ http://orcid.org/0000-0002-0462-6540

### Decision letter and Author response
Decision letter https://doi.org/10.7554/eLife.76494.sa1
Author response https://doi.org/10.7554/eLife.76494.sa2

## Additional files

### Supplementary files
• Supplementary file 1. Cryo-EM data processing, refinement and validation statistics (Supplementary Tables 1-25).
• Transparent reporting form
• Source data 1. Uncropped gel of *Figure 4—figure supplement 2*.

### Data availability
There are no restrictions on data and materials availability. Cryo-EM maps and atomic models have been deposited at the EMDB and the PDB, respectively (see Supplementary file 1 - Tables 1-25 for their accession codes). In addition, the raw cryo-EM data, together with the relevant intermediate steps of their processing have been deposited at EMPIAR for three data sets: EMPIAR-10940 for data set 11; EMPIAR-10943 for data set 10; EMPIAR-10944 for data set 15.

The following datasets were generated:

| Author(s) | Year | Dataset title | Dataset URL | Database and Identifier |
|---|---|---|---|---|
| Lovestam S, Scheres SHW | 2022 | Tutorial for high-throughput cryo-EM structure determination of amyloids: Dataset 1 | https://www.ebi.ac.uk/empiar/EMPIAR-10940 | EBI, 10940 |
| Lovestam S, Scheres SHW | 2022 | Tutorial for high-throughput cryo-EM structure determination of amyloids: Dataset 2 | https://www.ebi.ac.uk/empiar/EMPIAR-10943 | EBI, 10943 |
| Lovestam S, Scheres SHW | 2022 | Tutorial for high-throughput cryo-EM structure determination of amyloids: Dataset 3 | https://www.ebi.ac.uk/empiar/EMPIAR-10944 | EMPIAR, 10944 |
| Lovestam S, Scheres SHW | 2022 | In vitro assembled tau filaments into Quadruple Helical Filaments type 1 (2c) | https://www.rcsb.org/structure/7QJV | RCSB Protein Data Bank, 7QJV |
| Lovestam S, Scheres SHW | 2022 | In vitro assembled tau filaments into Quadruple Helical Filaments type 1 (2c) | https://www.ebi.ac.uk/emdb/EMD-14023 | Electron Microscopy Data Bank, 14023 |
| Lovestam S, Scheres SHW | 2022 | In vitro assembled tau filaments with structures like Paired Helical Filaments from Alzheimer's Disease | https://www.rcsb.org/structure/7QL4 | RCSB Protein Data Bank, 7QL4 |

*Continued on next page*

*Continued*

| Author(s) | Year | Dataset title | Dataset URL | Database and Identifier |
|---|---|---|---|---|
| Lovestam S, Scheres SHW | 2022 | In vitro assembled tau filaments with structures like Paired Helical Filaments from Alzheimer's Disease | https://www.ebi.ac.uk/emdb/EMD-14063 | Electron Microscopy Data Bank, 14063 |
| Lovestam S, Scheres SHW | 2022 | In vitro assembled tau filaments with structures like chronic traumatic encephalopathy type II (8a) | https://www.rcsb.org/structure/7QJW | RCSB Protein Data Bank, 7QJW |
| Lovestam S, Scheres SHW | 2022 | In vitro assembled tau filaments with structures like chronic traumatic encephalopathy type II (8a) | https://www.ebi.ac.uk/emdb/EMD-14024 | Electron Microscopy Data Bank, 14024 |
| Lovestam S, Scheres SHW | 2022 | In vitro assembled 266/297 - 391 tau filaments with NaCl (8b) | https://www.rcsb.org/structure/7QL3 | RCSB Protein Data Bank, 7QL3 |
| Lovestam S, Scheres SHW | 2022 | In vitro assembled 266/297 - 391 tau filaments with NaCl (8b) | https://www.ebi.ac.uk/emdb/EMD-14062 | Electron Microscopy Data Bank, 14062 |
| Lovestam S, Scheres SHW | 2022 | In vitro assembled 266/297 - 391 tau filaments with LiCl (9a) | https://www.rcsb.org/structure/7QJY | RCSB Protein Data Bank, 7QJY |
| Lovestam S, Scheres SHW | 2022 | In vitro assembled 266/297 - 391 tau filaments with LiCl (9a) | https://www.ebi.ac.uk/emdb/EMD-14026 | Electron Microscopy Data Bank, 14026 |
| Lovestam S, Scheres SHW | 2022 | In vitro assembled 266/297 - 391 tau filaments with LiCl (9b) | https://www.rcsb.org/structure/7QJZ | RCSB Protein Data Bank, 7QJZ |
| Lovestam S, Scheres SHW | 2022 | In vitro assembled 266/297 - 391 tau filaments with LiCl (9b) | https://www.ebi.ac.uk/emdb/EMD-14027 | Electron Microscopy Data Bank, 14027 |
| Lovestam S, Scheres SHW | 2022 | In vitro assembled 266/297 - 391 tau filaments with KCl (10a) | https://www.rcsb.org/structure/7QK5 | RCSB Protein Data Bank, 7QK5 |
| Lovestam S, Scheres SHW | 2022 | In vitro assembled 266/297 - 391 tau filaments with KCl (10a) | https://www.ebi.ac.uk/emdb/EMD-14038 | Electron Microscopy Data Bank, 14038 |
| Lovestam S, Scheres SHW | 2022 | In vitro assembled 266/297 - 391 tau filaments with KCl (10b) | https://www.rcsb.org/structure/7R5H | RCSB Protein Data Bank, 7R5H |
| Lovestam S, Scheres SHW | 2022 | In vitro assembled 266/297 - 391 tau filaments with KCl (10b) | https://www.ebi.ac.uk/emdb/EMD-14320 | Electron Microscopy Data Bank, 14320 |
| Lovestam S, Scheres SHW | 2022 | In vitro assembled 266/297 - 391 tau filaments with ZnCl2 (11a) | https://www.rcsb.org/structure/7QKL | RCSB Protein Data Bank, 7QKL |
| Lovestam S, Scheres SHW | 2022 | In vitro assembled 266/297 - 391 tau filaments with ZnCl2 (11a) | https://www.ebi.ac.uk/emdb/EMD-14046 | Electron Microscopy Data Bank, 14046 |
| Lovestam S, Scheres SHW | 2022 | In vitro assembled 266/297 - 391 tau filaments with CuCl2 (12a) | https://www.rcsb.org/structure/7QKF | RCSB Protein Data Bank, 7QKF |
| Lovestam S, Scheres SHW | 2022 | In vitro assembled 266/297 - 391 tau filaments with CuCl2 (12a) | https://www.ebi.ac.uk/emdb/EMD-14040 | Electron Microscopy Data Bank, 14040 |

*Continued*

| Author(s) | Year | Dataset title | Dataset URL | Database and Identifier |
|---|---|---|---|---|
| Lovestam S, Scheres SHW | 2022 | In vitro assembled 266/297 - 391 tau filaments with MgCl2 and NaCl (14a) | https://www.rcsb.org/structure/7QKU | RCSB Protein Data Bank, 7QKU |
| Lovestam S, Scheres SHW | 2022 | In vitro assembled 266/297 - 391 tau filaments with MgCl2 and NaCl (14a) | https://www.ebi.ac.uk/emdb/EMD-14053 | Electron Microscopy Data Bank, 14053 |
| Lovestam S, Scheres SHW | 2022 | In vitro assembled 266/297 - 391 tau filaments with MgCl2 and NaCl (14b) | https://www.rcsb.org/structure/7QKJ | RCSB Protein Data Bank, 7QKJ |
| Lovestam S, Scheres SHW | 2022 | In vitro assembled 266/297 - 391 tau filaments with MgCl2 and NaCl (14b) | https://www.ebi.ac.uk/emdb/EMD-14044 | Electron Microscopy Data Bank, 14044 |
| Lovestam S, Scheres SHW | 2022 | In vitro assembled tau filaments with MgSO4 and NaCl (15a) | https://www.rcsb.org/structure/7QKV | RCSB Protein Data Bank, 7QKV |
| Lovestam S, Scheres SHW | 2022 | In vitro assembled tau filaments with MgSO4 and NaCl (15a) | https://www.ebi.ac.uk/emdb/EMD-14054 | Electron Microscopy Data Bank, 14054 |
| Lovestam S, Scheres SHW | 2022 | In vitro assembled 266/297 - 391 tau filaments with MgSO4 and NaCl (15b) | https://www.rcsb.org/structure/7QKX | RCSB Protein Data Bank, 7QKX |
| Lovestam S, Scheres SHW | 2022 | In vitro assembled 266/297 - 391 tau filaments with MgSO4 and NaCl (15b) | https://www.ebi.ac.uk/emdb/EMD-14056 | Electron Microscopy Data Bank, 14056 |
| Lovestam S, Scheres SHW | 2022 | In vitro assembled 266/297 - 391 tau filaments with MgSO4 and NaCl (15c) | https://www.rcsb.org/structure/7QL0 | RCSB Protein Data Bank, 7QL0 |
| Lovestam S, Scheres SHW | 2022 | In vitro assembled 266/297 - 391 tau filaments with MgSO4 and NaCl (15c) | https://www.ebi.ac.uk/emdb/EMD-14059 | Electron Microscopy Data Bank, 14059 |
| Lovestam S, Scheres SHW | 2022 | In vitro assembled 266/297 - 391 tau filaments with NaHCO3 and NaCl (16a) | https://www.rcsb.org/structure/7R4T | RCSB Protein Data Bank, 7R4T |
| Lovestam S, Scheres SHW | 2022 | In vitro assembled 266/297 - 391 tau filaments with NaHCO3 and NaCl (16a) | https://www.ebi.ac.uk/emdb/EMD-14316 | Electron Microscopy Data Bank, 14316 |
| Lovestam S, Scheres SHW | 2022 | In vitro assembled 244-391 tau filaments with Na2P2O7 (20a) | https://www.rcsb.org/structure/7QL2 | RCSB Protein Data Bank, 7QL2 |
| Lovestam S, Scheres SHW | 2022 | In vitro assembled 244-391 tau filaments with Na2P2O7 (20a) | https://www.ebi.ac.uk/emdb/EMD-14061 | Electron Microscopy Data Bank, 14061 |
| Lovestam S, Scheres SHW | 2022 | In vitro assembled 266-391 tau filaments in PBS (23a) | https://www.rcsb.org/structure/7QL1 | RCSB Protein Data Bank, 7QL1 |
| Lovestam S, Scheres SHW | 2022 | In vitro assembled 266-391 tau filaments in PBS (23a) | https://www.ebi.ac.uk/emdb/EMD-14060 | Electron Microscopy Data Bank, 14060 |
| Lovestam S, Scheres SHW | 2022 | In vitro assembled 305-379 tau filaments (27a) | https://www.rcsb.org/structure/7QKZ | RCSB Protein Data Bank, 7QKZ |
| Lovestam S, Scheres SHW | 2022 | In vitro assembled 305-379 tau filaments (27a) | https://www.ebi.ac.uk/emdb/EMD-14058 | Electron Microscopy Data Bank, 14058 |
| Lovestam S, Scheres SHW | 2022 | In vitro assembled 297-394 tau filaments, 700 rpm (34a) | https://www.rcsb.org/structure/7QJX | RCSB Protein Data Bank, 7QJX |

*Continued on next page*

*Continued*

| Author(s) | Year | Dataset title | Dataset URL | Database and Identifier |
|---|---|---|---|---|
| Lovestam S, Scheres SHW | 2022 | In vitro assembled 297-394 tau filaments, 700 rpm (34a) | https://www.ebi.ac.uk/emdb/EMD-14025 | Electron Microscopy Data Bank, 14025 |
| Lovestam S, Scheres SHW | 2022 | In vitro assembled 297-394 tau filaments in PBS (35d) | https://www.rcsb.org/structure/7QK1 | RCSB Protein Data Bank, 7QK1 |
| Lovestam S, Scheres SHW | 2022 | In vitro assembled 297-394 tau filaments in PBS (35d) | https://www.ebi.ac.uk/emdb/EMD-14028 | Electron Microscopy Data Bank, 14028 |
| Lovestam S, Scheres SHW | 2022 | In vitro assembled 300-391 tau filaments in PBS (36a) | https://www.rcsb.org/structure/7QK2 | RCSB Protein Data Bank, 7QK2 |
| Lovestam S, Scheres SHW | 2022 | In vitro assembled 300-391 tau filaments in PBS (36a) | https://www.ebi.ac.uk/emdb/EMD-14029 | Electron Microscopy Data Bank, 14029 |
| Lovestam S, Scheres SHW | 2022 | In vitro assembled 258-391 tau filaments, 700 rpm (38a) | https://www.rcsb.org/structure/7QK3 | RCSB Protein Data Bank, 7QK3 |
| Lovestam S, Scheres SHW | 2022 | In vitro assembled 258-391 tau filaments, 700 rpm (38a) | https://www.ebi.ac.uk/emdb/EMD-14030 | Electron Microscopy Data Bank, 14030 |
| Lovestam S, Scheres SHW | 2022 | In vitro assembled 258-391 tau filaments with phosphoglycerate, 700 rpm (39a) | https://www.rcsb.org/structure/7QKG | RCSB Protein Data Bank, 7QKG |
| Lovestam S, Scheres SHW | 2022 | In vitro assembled 258-391 tau filaments with phosphoglycerate, 700 rpm (39a) | https://www.ebi.ac.uk/emdb/EMD-14041 | Electron Microscopy Data Bank, 14041 |
| Lovestam S, Scheres SHW | 2022 | In vitro assembled 258-391 tau filaments with heparan sulphate, 700 rpm (40a) | https://www.rcsb.org/structure/7QK6 | RCSB Protein Data Bank, 7QK6 |
| Lovestam S, Scheres SHW | 2022 | In vitro assembled 258-391 tau filaments with heparan sulphate, 700 rpm (40a) | https://www.ebi.ac.uk/emdb/EMD-14039 | Electron Microscopy Data Bank, 14039 |
| Lovestam S, Scheres SHW | 2022 | In vitro assembled 258-391 tau filaments with sodium azide, (41a) | https://www.rcsb.org/structure/7QKH | RCSB Protein Data Bank, 7QKH |
| Lovestam S, Scheres SHW | 2022 | In vitro assembled 258-391 tau filaments with sodium azide, (41a) | https://www.ebi.ac.uk/emdb/EMD-14042 | Electron Microscopy Data Bank, 14042 |
| Lovestam S, Scheres SHW | 2022 | In vitro assembled 297-408 S396D S400D T403D S404D tau filaments (42a) | https://www.rcsb.org/structure/7QKI | RCSB Protein Data Bank, 7QKI |
| Lovestam S, Scheres SHW | 2022 | In vitro assembled 297-408 S396D S400D T403D S404D tau filaments (42a) | https://www.ebi.ac.uk/emdb/EMD-14043 | Electron Microscopy Data Bank, 14043 |
| Lovestam S, Scheres SHW | 2022 | In vitro assembled 297-441 S396D S400D T403D S404D tau filaments (43a) | https://www.rcsb.org/structure/7QKK | RCSB Protein Data Bank, 7QKK |
| Lovestam S, Scheres SHW | 2022 | In vitro assembled 297-441 S396D S400D T403D S404D tau filaments (43a) | https://www.ebi.ac.uk/emdb/EMD-14045 | Electron Microscopy Data Bank, 14045 |
| Lovestam S, Scheres SHW | 2022 | In vitro assembled 266-391 S356D tau filaments with KCl (44a) | https://www.rcsb.org/structure/7QKW | RCSB Protein Data Bank, 7QKW |
| Lovestam S, Scheres SHW | 2022 | In vitro assembled 266-391 S356D tau filaments with KCl (44a) | https://www.ebi.ac.uk/emdb/EMD-14055 | Electron Microscopy Data Bank, 14055 |

*Continued on next page*

*Continued*

| Author(s) | Year | Dataset title | Dataset URL | Database and Identifier |
|---|---|---|---|---|
| Lovestam S, Scheres SHW | 2022 | In vitro assembled 266-391 S356D tau filaments with NaCl (45a) | https://www.rcsb.org/structure/7QKM | RCSB Protein Data Bank, 7QKM |
| Lovestam S, Scheres SHW | 2022 | In vitro assembled 266-391 S356D tau filaments with NaCl (45a) | https://www.ebi.ac.uk/emdb/EMD-14047 | Electron Microscopy Data Bank, 14047 |
| Lovestam S, Scheres SHW | 2022 | In vitro assembled 0N4R tau filaments with phosphoserine (47a) | https://www.rcsb.org/structure/7QKY | RCSB Protein Data Bank, 7QKY |
| Lovestam S, Scheres SHW | 2022 | In vitro assembled 0N4R tau filaments with phosphoserine (47a) | https://www.ebi.ac.uk/emdb/EMD-14057 | Electron Microscopy Data Bank, 14057 |

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
