## [Editor Report]

In this paper 76 cryo-EM structures of recombinant tau filaments assembled in vitro are described. This is a scientific tour-de-force, and will provide an immense database that can be used by everyone working in the amyloid field. When this knowledge is combined with the structure of tau filaments in vivo, it will help shape the design of laboratory experiments in the future to generate the conditions to replicate the in vivo forms in vitro.

---

## [Decision Letter]

**Decision letter after peer review:**

Thank you for submitting your article "Assembly of recombinant tau into filaments identical to those of Alzheimer's disease and chronic traumatic encephalopathy" for consideration by *eLife*. Your article has been reviewed by 4 peer reviewers, including Edward H Egelman as the Reviewing Editor and Reviewer #1, and the evaluation has been overseen by Jeannie Chin as the Senior Editor. The following individuals involved in review of your submission have agreed to reveal their identity: Fengbin Wang (Reviewer #3); Louise C Serpell (Reviewer #4).

Essential revisions:

1) For one filament solved at 1.8 Å resolution, they determined the hand of these polymers from the carbonyl oxygens. But it is unclear how the hand was determined or chosen for the rest. The reason that this is important is because it is stated: "Whereas all previously described tau filaments had a left-handed twist, phosphoserine-induced filaments were right-handed." Is it actually true that all previously described tau filaments had a left-handed twist, or was that assumed in many papers? The authors of two recent papers in the amyloid field were asked about what experimental evidence they had for stating that their filaments were left-handed. In both cases, they replied that this was an assumption based upon the literature, and it was a mistake to never state this in the paper (or, in one case, to actually imply that the filaments were observed to be left-handed). While it might be thought that one might require almost near-atomic resolution to see the hand directly, a recent PNAS paper on a largely β structure (10.1073/pnas.2120346119) showed that the hand could be determined at 2.5 Å resolution, and a Nature Communications paper (10.1038/s41467-020-19512-3) showed this at 2.6 Å resolution. It would help the amyloid field greatly to explicitly state in all cases when the hand has been determined, when it has been assumed, and test what resolution might be required to have confidence in describing the hand.

2) There is a basic question about cryo-EM studies of fibrils that should be addressed directly and honestly by the authors, namely that the cryo-EM densities and molecular models are generally constructed from less than half of the particles that are originally picked, in some cases much less than half (e.g., 12% for polymorph "2c", a combined 14% for polymorphs "8a" and "8b", 5% for polymorph "10a", etc., according to the supplementary tables). In addition, much of the fibrillar material that is prepared may be lost in the initial centrifugation step to "remove any large aggregates", and fibrils that are self-associated on the cryo-EM grids are probably not picked. Therefore, it seems to me that statements to the effect that certain conditions do or do not produce certain polymorphs (or predominantly produce certain polymorphs) can not be strongly justified. The authors should admit that this is a real problem, explain how they deal with this problem, and include appropriate caveats and modifications to their language, to clarify that their conclusions are based entirely on the fibrils that are observable as distinct entities by cryo-EM, which could potentially be a subset of the fibrils that are present in the original samples. (The same statements apply to cryo-EM of brain-extracted fibrils as well, although that is not the subject of this manuscript.) How can one have 100% unless entire grids have been examined?

3. Table 1 should include columns that summarize the "folds" that are observed under the various sets of conditions. For example, "1a" is a new fold, while "2a", "3a", "4a", etc. are the AD PHF fold. Including this information in Table 1 would make the paper much easier to follow.

4. I strongly recommend that the authors develop and include a quantitative measure of the similarity of one cryo-EM density to another, and/or one structural model to another. Currently, they are using rather qualitative and potentially subjective visual comparisons or overlays, but they are making strong statements such as "…the first type was IDENTICAL to CTE type II filaments". For structural models, they could use the rmsds of heavy atoms. For densities, I suppose they could use an approach related to the FSC method that is used to evaluate the resolution of a single density. (Perhaps a good method already exists, but I am not aware of it.)

5. What do the authors mean by "high-throughput cryo-EM structure determination"? This should be defined or explained in the main text.

6. Have the authors already tried to propagate the structures of their brain-extracted tau fibrils in vitro, by seeded fibril growth as in the experiments on amyloid-β fibrils by the Tycko lab? I know that the MRC group has published a paper claiming that brain-extracted α-synuclein fibrils can not be propagated in vitro (at least not with the specific approach used in their FEBS OpenBio paper). If they have tried to propagate brain-extracted tau fibrils in vitro, they should mention this in the current manuscript. If not, they should consider performing such experiments and including the results in this manuscript. For example, it would be interesting to know whether the in vitro growth conditions that result in AD PHF polymorphs are also good conditions for propagation of genuine PHFs from AD brain tissue, while other conditions are not as good.

7. The authors say "we previously showed that heparin-induced tau filament formation leads to structures that are different from those present in disease". This is one example of an overly confident statement, related to the points raised above. In light of the strong dependence of tau fibril structures on the precise details of growth conditions indicated by the results in this manuscript, it seems quite possible, even likely, that conditions can be found under which heparin-induced tau filaments are the SAME as those present in disease. The authors should modify this statement, and others like it throughout the paper, to indicate that this statement is based on a specific set of experiments, performed under specific conditions, leaving open the possibility that other conditions could produce different results.

8. The authors say "filaments were picked manually or automatically using Topaz". This needs clarification. In particular, how useful was the Topaz approach? How many of the reported densities are based on automatic picking? Or for a given density, what fraction of the particles resulted from automatic vs. manual picking? Did the automatically-picked filaments need to be manually curated before being used? When was Topaz successful, and when was it not?

9. The authors' choices of references at certain points in the manuscript seem unjustified. For example, the 2011 paper by Fitzpatrick et al. is not an appropriate reference for the statement that "amyloid filament formation is driven by hydrogen bond formation between the extended β-sheets along the helical axis", as this was a well-established fact about amyloid structures long before 2011 (e.g., from x-ray fiber diffraction and solid state NMR). Similarly, references for the statement that "multiple different amyloid structures have been reported for proteins like tau, amyloid-β, α-synuclein,…" are limited to the relatively recent papers based on cryo-EM measurements, when in fact polymorphism was demonstrated in these cases by solid state NMR much earlier, and molecular structural models for amyloid-β and α-synuclein were developed years before the first truly successful cryo-EM results.

10. Along similar lines, I am not sure what to make of the statement that "For many years, amyloid formation has been studied…Until recently, these studies were blind to the structures being formed." This statement seems to ignore a large set of structural studies of α-synuclein, amyloid-β, huntingtin, tau, and other fibrils prior to 2017.

11. The statement that "it is possible that the cavity in the filaments formed with NaCl is FILLED with Na^+^ and Cl^-^ ions" seems rather far-fetched, perhaps because I do not understand what the authors mean. What density of Na^+^ and Cl^-^ would they need to explain their data?

12. The statement that "even distance restraints derived from solid-state NMR are probably not sufficient to tell all the different structures apart" is not relevant. It is well established from solid state NMR studies of tau, α-synuclein, and amyloid-β fibrils that different polymorphs are readily distinguished by differences in NMR chemical shifts, which result in different crosspeak patterns in 2D or 3D spectra. These crosspeak patterns are highly sensitive "fingerprints" of individual polymorphs, and are easy to obtain with 15N,13C-labeled samples. Distance restraints are used to develop full structural models, but are not needed to distinguish one polymorph from another. One can screen the dependence of various polymorph populations on growth conditions by recording and comparing the 2D or 3D spectra, which might take 0.5-5 days per sample, depending on the available quantities. The only issue then is whether the brain-extracted fibril structures can be self-propagated once by seeded growth, to create 15N,13C-labeled fibrils whose 2D or 3D spectra can be compared with spectra of fibrils that are grown entirely in vitro.

13. In the introduction (p2) the authors state that the addition of RNA or Phosphoserine to full length 4R tau lead to filament structures "different to those observed in disease". I think it is important to say "solved thus far". Of course, the research group have solved a very large number of disease related structure, but they can not rule out further structures that will be solved in the future and related to disease. I think this is a point to consider through out. I agree these are "new" structures, but they are not necessarily "non-physiological". It maybe that these structures have not yet been discovered ex-vivo (or in vivo).

14. On page 3, the authors mention some previous studies of dGAE and these were conducted with atomic force microscopy analysis as well as negative stain electron microscopy. So this should be updated. It could be useful to mention that one of these studies included comparison with sectioned AD tissue where the morphology of PHF could be observed (where filaments had not been extracted).

15. On page 4, the authors talk about THFs and QHFs that "have not been observed from AD brain". CryoEM provides structures for the most highly populated polymorph that can be extracted and averaged and so it is possible that THFs and QHFs could form in vivo or be present in vivo in minor proportions of the population. The author can not unequivocally discount their presence in vivo so I suggest reducing the tone of these sentences here and throughout.

16. The authors describe the structures of filaments formed in the "conditions of Al-Hilaly" (pages 5 and 6 and others). Firstly, this is not really helpful for the general reader and will make far more sense to replace this throughout with the conditions used (PB, pH7.4 etc). Furthermore, it is not possible for the authors to be certain that the conditions are identical to those used by other researchers, particularly given that the source of purified protein is different and the process of purification differs. The source of protein is likely to be of key importance for the generation of different polymorphs and will effect the morphologies formed. The only way to be sure that those formed under conditions used in Al-Hilaly et al., would be to use protein from the original source under identical conditions. In a wider context, this paper provides really important and useful information regarding the structures of tau filaments formed under varying conditions, but another researcher may not be able to reproduce these under the same conditions due to the nature of the purification variation as well as more subtle differences in the water, temperature, time frames and so on, even the shaker and the tubes or plates used will have an influence on the morphologies formed. This influence of small variations is well known for amyloid in general and it is important that the authors do not claim to know exactly how to make an AD PHF in general – but they do know how to make it under the conditions they use in their environment using their source of purified protein.

17. A general question pertaining to the morphologies observed : how exactly are particles selected for processing and then how does a class or classes get picked for further processing. Is it possible (in principle) to miss a population which has a low representation?

18. are the authors able to confirm that full length tau remains full length throughout the assembly process? It is possible (likely?) that a degree of degradation/fragmentation may occur leading to loss of regions of the protein and resulting in the composition of the filament being formed of shorter fragments? Could this be ruled out (e.g. immunolabelling to N/C termini; SDS page showing single protein without degradation products). It is really interesting that introduction of Asp for Ser converts the full-length protein to an aggregation prone protein. Does the sample condition include any protease inhibitors?

19. In the RNA enhanced structure, is any density available for RNA? I don't believe an image of the structure is provided in the paper or supplementary although I may have missed it

20. Overall, the paper is incredibly densely packed with structures which are really exciting. I wonder if the authors could consider additional labelling to help readers orient themselves around the structures. Previous work from this group has nicely included figures with labelled residues (in circles) which helps to examine these structures. For example, the authors have talked about a flip of the proline in one region but it is very difficult to see this in these quite small figures. I wondered whether there is a trans-isomerisation of the proline which might influence different morphology formation.

I think the paper would be even further enhanced by additional labelling to put structures in clearer context.

21. top of page 3 there is a. missing

GGT is referred to but it has not been defined or previously mentioned

22. Some of the structures are missing, such as 46, which is important since it is an RNA structure.

23. There appears to be a mistake when structure 11a is referred to as being in KCl and with C3 symmetry.

---

## [Author Response]

Essential revisions:1) For one filament solved at 1.8 Å resolution, they determined the hand of these polymers from the carbonyl oxygens. But it is unclear how the hand was determined or chosen for the rest. The reason that this is important is because it is stated: "Whereas all previously described tau filaments had a left-handed twist, phosphoserine-induced filaments were right-handed." Is it actually true that all previously described tau filaments had a left-handed twist, or was that assumed in many papers? The authors of two recent papers in the amyloid field were asked about what experimental evidence they had for stating that their filaments were left-handed. In both cases, they replied that this was an assumption based upon the literature, and it was a mistake to never state this in the paper (or, in one case, to actually imply that the filaments were observed to be left-handed). While it might be thought that one might require almost near-atomic resolution to see the hand directly, a recent PNAS paper on a largely β structure (10.1073/pnas.2120346119) showed that the hand could be determined at 2.5 Å resolution, and a Nature Communications paper (10.1038/s41467-020-19512-3) showed this at 2.6 Å resolution. It would help the amyloid field greatly to explicitly state in all cases when the hand has been determined, when it has been assumed, and test what resolution might be required to have confidence in describing the hand.

At resolutions higher than 2.0 Å, the hand of an amyloid filament can be derived directly from the chirality of individual amino acid residues. At lower resolutions, the hand can be deduced from chiral features of secondary structure elements. In the β-strands, these include the orientations of main chain carbonyl groups relative to the side chains. Small bumps in the electron density maps, corresponding to the main chain oxygen atoms, are discernable at resolutions beyond ~2.9 Å. Cross-β amyloids are devoid of α-helices, but they may contain polyproline-like left-handed helices. There are such helices in some Tau filament structures, in particular in the protofilament interfaces of PHF AD and Type II CTE, formed by the PGGG motif at the end of R3. There is a new occurrence of such helices, formed by the PGGG motif at the end of R2, in the protofilament interface of structure 41a, solved at 3.2Å resolution. In the left-handed map for 41a filaments, these helices are left-handed.

We have added the following paragraph to the Methods section to clarify how the handedness of different filament types was determined:

“For reconstructions with estimated resolutions beyond 2.9 Å, densities for the backbone oxygen atoms provide direct evidence for the handedness of the filament. In this manner, both AD and CTE filaments were shown to be left-handed based (Falcon et al., 2019; Shi, et al., 2021). […] The handedness of filaments with previously unobserved protofilament folds that were solved at resolutions insufficient for atomic modelling (i.e. filament types 19a, 28a, 29a, 30a, 31a and 46a) remains unclear.”

We have also modified the statement about all previously observed tau filaments being left-handed. This sentence now reads:

“Whereas most tau filaments described in this paper have a left-handed twist (see Methods), phosphoserine-induced filaments are right-handed.”

2) There is a basic question about cryo-EM studies of fibrils that should be addressed directly and honestly by the authors, namely that the cryo-EM densities and molecular models are generally constructed from less than half of the particles that are originally picked, in some cases much less than half (e.g., 12% for polymorph "2c", a combined 14% for polymorphs "8a" and "8b", 5% for polymorph "10a", etc., according to the supplementary tables). In addition, much of the fibrillar material that is prepared may be lost in the initial centrifugation step to "remove any large aggregates", and fibrils that are self-associated on the cryo-EM grids are probably not picked. Therefore, it seems to me that statements to the effect that certain conditions do or do not produce certain polymorphs (or predominantly produce certain polymorphs) can not be strongly justified. The authors should admit that this is a real problem, explain how they deal with this problem, and include appropriate caveats and modifications to their language, to clarify that their conclusions are based entirely on the fibrils that are observable as distinct entities by cryo-EM, which could potentially be a subset of the fibrils that are present in the original samples. (The same statements apply to cryo-EM of brain-extracted fibrils as well, although that is not the subject of this manuscript.) How can one have 100% unless entire grids have been examined?

There are two aspects to this point. First, there is the selection of a subset of particles from a larger set of initially picked particles to calculate the final reconstruction. This is common practice in cryo-EM structure determination. It is understood to be an efficient way to improve the quality and resolution of cryo-EM reconstructions. Noise artefacts in the images, neighbouring particles, damaged particles, variations in ice thickness and probably many more factors lead to a variety in the signal-to-noise ratios and alignability of individual particles. Thanks to powerful image classification algorithms, the best particles can be identified and used for final reconstruction. Using more particles typically leads to the same structures, but at the expense of the quality and resolution of the reconstruction.

Second, there is the argument that certain filament types may clump together more than others and thereby be removed during sample preparation (or image processing). We note that sample preparation artefacts are relevant for all techniques in structural biology and that they are generally hard to address. We do often observe large clumps of filaments on cryo-EM grids (also for ex vivo samples). This was for example the case for dataset 2 (a-d) and required imaging at the periphery of the clumps to obtain sufficient data. We typically find that the structures at the periphery of clumps are the same as those from other parts of the grid. The centrifuge spin prior to grid plunging helps to remove the majority of large clumps, which would otherwise stick to the grid and lead to ice that is too thick for imaging.

We have added the following statement to the Methods section:

“Reported percentages of filament types were calculated from the number of assigned segments to 2D classes. Filament types 15c and 15d could not be distinguished from their 2D class averages. In this case, fuzzy density in the reconstruction hinted at remaining structural heterogeneity in the data. These filament types were separated by 3D classification, and the reported percentages were calculated from the number of segments in the 3D classes. We note that experimental noise in the cryo-EM images leads to stochasticity in their class assignments, and that sample preparation artefacts may affect different filament types to different extents. Therefore, the reported percentages might not reflect the exact relative amounts of filament types in the original assembly reaction.”

3. Table 1 should include columns that summarize the "folds" that are observed under the various sets of conditions. For example, "1a" is a new fold, while "2a", "3a", "4a", etc. are the AD PHF fold. Including this information in Table 1 would make the paper much easier to follow.

We have updated Table 1 as suggested.

4. I strongly recommend that the authors develop and include a quantitative measure of the similarity of one cryo-EM density to another, and/or one structural model to another. Currently, they are using rather qualitative and potentially subjective visual comparisons or overlays, but they are making strong statements such as "…the first type was IDENTICAL to CTE type II filaments". For structural models, they could use the rmsds of heavy atoms. For densities, I suppose they could use an approach related to the FSC method that is used to evaluate the resolution of a single density. (Perhaps a good method already exists, but I am not aware of it.)

At various places in the revised manuscript, we have inserted all-atom (except hydrogens) r.m.s.d. values when we claim structures are (nearly) identical to others.

5. What do the authors mean by "high-throughput cryo-EM structure determination"? This should be defined or explained in the main text.

We now define “high-throughput cryo-EM structure determination” in the Discussion:

“Whereas solving a cryo-EM structure previously constituted a scientific project in itself, current throughputs allow the use of cryo-EM structure determination as a tool in projects with a wider scope. We envision that this type of high-throughput cryo-EM structure determination will play a crucial role in future investigations into the structural diversity of amyloids.”

6. Have the authors already tried to propagate the structures of their brain-extracted tau fibrils in vitro, by seeded fibril growth as in the experiments on amyloid-β fibrils by the Tycko lab? I know that the MRC group has published a paper claiming that brain-extracted α-synuclein fibrils can not be propagated in vitro (at least not with the specific approach used in their FEBS OpenBio paper). If they have tried to propagate brain-extracted tau fibrils in vitro, they should mention this in the current manuscript. If not, they should consider performing such experiments and including the results in this manuscript. For example, it would be interesting to know whether the in vitro growth conditions that result in AD PHF polymorphs are also good conditions for propagation of genuine PHFs from AD brain tissue, while other conditions are not as good.

This is one of the exciting opportunities that our in vitro assembled filaments bring for further research, and we certainly have plans to perform experiments along the lines suggested. However, because this paper is already long and filled with a large amount of experimental data, we suggest such experiments would be better in a separate, future paper.

7. The authors say "we previously showed that heparin-induced tau filament formation leads to structures that are different from those present in disease". This is one example of an overly confident statement, related to the points raised above. In light of the strong dependence of tau fibril structures on the precise details of growth conditions indicated by the results in this manuscript, it seems quite possible, even likely, that conditions can be found under which heparin-induced tau filaments are the SAME as those present in disease. The authors should modify this statement, and others like it throughout the paper, to indicate that this statement is based on a specific set of experiments, performed under specific conditions, leaving open the possibility that other conditions could produce different results.

We have replaced this statement, and similar ones elsewhere, with:

“We previously showed that heparin-induced tau filament formation led to structures that are different from those observed in disease thus far, under the conditions used”.

8. The authors say "filaments were picked manually or automatically using Topaz". This needs clarification. In particular, how useful was the Topaz approach? How many of the reported densities are based on automatic picking? Or for a given density, what fraction of the particles resulted from automatic vs. manual picking? Did the automatically-picked filaments need to be manually curated before being used? When was Topaz successful, and when was it not?

Entire data sets were picked either manually or using Topaz. We have added the following sentences to the Methods section:

“In total, 38 data sets were picked using Topaz. Data sets with low contrast, due to thick ice or imaging too close to focus, required manual picking.”

In addition, aiming at a wide adaptation of our methods in the field, we have now written a methods-oriented paper that describes the Topaz implementation, as well as general guidelines about amyloid structure determination, in more detail. A preprint of this paper is available on bioRxiv (DOI: 10.1101/2022.02.07.479378), and we now cite this preprint in the Methods section.

As per instructions of the editorial team, we have also included a reference to the commit on github, where we host the modified Topaz code.

9. The authors' choices of references at certain points in the manuscript seem unjustified. For example, the 2011 paper by Fitzpatrick et al. is not an appropriate reference for the statement that "amyloid filament formation is driven by hydrogen bond formation between the extended β-sheets along the helical axis", as this was a well-established fact about amyloid structures long before 2011 (e.g., from x-ray fiber diffraction and solid state NMR). Similarly, references for the statement that "multiple different amyloid structures have been reported for proteins like tau, amyloid-β, α-synuclein,…" are limited to the relatively recent papers based on cryo-EM measurements, when in fact polymorphism was demonstrated in these cases by solid state NMR much earlier, and molecular structural models for amyloid-β and α-synuclein were developed years before the first truly successful cryo-EM results.

We have removed the sentence with the Fitzpatrick citation, and have added older citations to ssNMR structures for several of the other proteins. The revised text reads:

“The observation that the same proteins can adopt multiple different amyloid structures has made it clear that the paradigm by which a protein's sequence defines its structure may not hold for amyloids. Apparently, the packing of β-sheets against each other in amyloid filaments can happen in many ways for a single amino acid sequence. Our work described here and performed previously (Zhang et al., 2019; Shi, Zhang, et al., 2021) illustrates this for tau. Similar observations have been made for amyloid-β (Bertini et al., 2011; Lu et al., 2013; Xiao et al., 2015, p. 201; Wälti et al., 2016; Kollmer et al., 2019; Yang et al., 2022), α-synuclein (Tuttle et al., 2016; B. Li et al., 2018; Y. Li et al., 2018; Guerrero-Ferreira et al., 2019; Schweighauser et al., 2020; Lövestam et al., 2021), TAR DNA binding protein-43 (TDP-43) (Arseni et al., 2021; Li, Babinchak and Surewicz, 2021) and immunoglobulin light chain (Radamaker et al., 2019; Swuec et al., 2019).”

10. Along similar lines, I am not sure what to make of the statement that "For many years, amyloid formation has been studied…Until recently, these studies were blind to the structures being formed." This statement seems to ignore a large set of structural studies of α-synuclein, amyloid-β, huntingtin, tau, and other fibrils prior to 2017.

We have removed this statement from the manuscript.

11. The statement that "it is possible that the cavity in the filaments formed with NaCl is FILLED with Na^+^ and Cl^-^ ions" seems rather far-fetched, perhaps because I do not understand what the authors mean. What density of Na^+^ and Cl^-^ would they need to explain their data?

We have changed the sentence to:

“Therefore, it is possible that the extra density in the cavity of filaments formed with NaCl also corresponds to Na^+^ and Cl^-^ ions”.

At this point we are not fully confident that the extra density in the cavity of in vitro filaments formed NaCl or CTE filaments corresponds to Na^+^ and Cl^-^ ions. However, as this hypothesis could guide important follow-up experiments, we considered it important to spell out and have used *eLife*’s new section on “Ideas and speculations” to do so***.***

12. The statement that "even distance restraints derived from solid-state NMR are probably not sufficient to tell all the different structures apart" is not relevant. It is well established from solid state NMR studies of tau, α-synuclein, and amyloid-β fibrils that different polymorphs are readily distinguished by differences in NMR chemical shifts, which result in different crosspeak patterns in 2D or 3D spectra. These crosspeak patterns are highly sensitive "fingerprints" of individual polymorphs, and are easy to obtain with 15N,13C-labeled samples. Distance restraints are used to develop full structural models, but are not needed to distinguish one polymorph from another. One can screen the dependence of various polymorph populations on growth conditions by recording and comparing the 2D or 3D spectra, which might take 0.5-5 days per sample, depending on the available quantities. The only issue then is whether the brain-extracted fibril structures can be self-propagated once by seeded growth, to create 15N,13C-labeled fibrils whose 2D or 3D spectra can be compared with spectra of fibrils that are grown entirely in vitro.

We have removed the statement about NMR distance restraints from the manuscript.

13. In the introduction (p2) the authors state that the addition of RNA or Phosphoserine to full length 4R tau lead to filament structures "different to those observed in disease". I think it is important to say "solved thus far". Of course, the research group have solved a very large number of disease related structure, but they can not rule out further structures that will be solved in the future and related to disease. I think this is a point to consider through out. I agree these are "new" structures, but they are not necessarily "non-physiological". It maybe that these structures have not yet been discovered ex-vivo (or in vivo).

We have changed the sentence as suggested. It now reads:

“Below, we show that the addition of RNA or phosphoserine to full-length recombinant 4R tau also led to filament structures that are different from those observed in disease thus far.”

14. On page 3, the authors mention some previous studies of dGAE and these were conducted with atomic force microscopy analysis as well as negative stain electron microscopy. So this should be updated. It could be useful to mention that one of these studies included comparison with sectioned AD tissue where the morphology of PHF could be observed (where filaments had not been extracted).

We have updated the Lutter citation and added ‘atomic force microscopy’ to this sentence. It now reads:

“It assembles into filaments with similar morphologies to AD PHFs by negative staining EM and atomic force microscopy (Novak, Kabat and Wischik, 1993; Al-Hilaly et al., 2017, 2020; Lutter et al., 2022).”

15. On page 4, the authors talk about THFs and QHFs that "have not been observed from AD brain". CryoEM provides structures for the most highly populated polymorph that can be extracted and averaged and so it is possible that THFs and QHFs could form in vivo or be present in vivo in minor proportions of the population. The author can not unequivocally discount their presence in vivo so I suggest reducing the tone of these sentences here and throughout.

No Tau filaments consisting of more than two protofilaments have been observed in a dozen or so tauopathies, including AD, studied ex vivo by cryo-EM thus far. Possibly, this is because their formation would leave less room available for the accommodation of large unstructured parts of the full-length tau chains.

Besides the issue about certain filament types potentially being lost due to sample preparation artefacts that was addressed under point 2, an additional question about filament types remaining undetected in the image processing is raised. We note that filament types from minority populations as small as 2% have been reported in this work and elsewhere (e.g. Falcon et al. 2019 and Shi et al., Nature 2021), and that larger differences between filament types make it easier to detect them. Because THFs and QHFs consist of one or two extra protofilaments compared to PHFs and SFs, they should be easy to detect in AD brain preps. Therefore, we are relatively confident that we would have observed these, if they had represented at least a few percent of the filaments in AD brain.

In any case, the statement “THFs and QHFs have not been observed in brain extracts from individuals with AD” remains accurate.

16. The authors describe the structures of filaments formed in the "conditions of Al-Hilaly" (pages 5 and 6 and others). Firstly, this is not really helpful for the general reader and will make far more sense to replace this throughout with the conditions used (PB, pH7.4 etc). Furthermore, it is not possible for the authors to be certain that the conditions are identical to those used by other researchers, particularly given that the source of purified protein is different and the process of purification differs. The source of protein is likely to be of key importance for the generation of different polymorphs and will effect the morphologies formed. The only way to be sure that those formed under conditions used in Al-Hilaly et al., would be to use protein from the original source under identical conditions. In a wider context, this paper provides really important and useful information regarding the structures of tau filaments formed under varying conditions, but another researcher may not be able to reproduce these under the same conditions due to the nature of the purification variation as well as more subtle differences in the water, temperature, time frames and so on, even the shaker and the tubes or plates used will have an influence on the morphologies formed. This influence of small variations is well known for amyloid in general and it is important that the authors do not claim to know exactly how to make an AD PHF in general – but they do know how to make it under the conditions they use in their environment using their source of purified protein.

We now refer to this condition as condition 1 (Table 1).

17. A general question pertaining to the morphologies observed : how exactly are particles selected for processing and then how does a class or classes get picked for further processing. Is it possible (in principle) to miss a population which has a low representation?

See our replies to points 2 and 15. Detailed instructions and examples of how to select classes are included in the preprint that is also mentioned under point 8.

18. are the authors able to confirm that full length tau remains full length throughout the assembly process? It is possible (likely?) that a degree of degradation/fragmentation may occur leading to loss of regions of the protein and resulting in the composition of the filament being formed of shorter fragments? Could this be ruled out (e.g. immunolabelling to N/C termini; SDS page showing single protein without degradation products). It is really interesting that introduction of Asp for Ser converts the full-length protein to an aggregation prone protein. Does the sample condition include any protease inhibitors?

Following the reviewer’s suggestion, we have checked the filament assembly reactions of the phosphomimetic mutants and found no evidence for proteolysis. We have inserted an image of the corresponding SDS-PAGE gel as Figure 4 —figure supplement 2. The corresponding text in the Results section reads:

“Analysis by gel electrophoresis confirmed that the tau constructs forming these filaments remained intact (Figure 4 —figure supplement 2).”

19. In the RNA enhanced structure, is any density available for RNA? I don't believe an image of the structure is provided in the paper or supplementary although I may have missed it

This structure was not solved to sufficient resolution for atomic modelling and is therefore only present in Figure 1 —figure supplement 1 (number 46a). Without atomic modelling, we feel not comfortable making claims about the presence of RNA density.

20. Overall, the paper is incredibly densely packed with structures which are really exciting. I wonder if the authors could consider additional labelling to help readers orient themselves around the structures. Previous work from this group has nicely included figures with labelled residues (in circles) which helps to examine these structures. For example, the authors have talked about a flip of the proline in one region but it is very difficult to see this in these quite small figures. I wondered whether there is a trans-isomerisation of the proline which might influence different morphology formation.I think the paper would be even further enhanced by additional labelling to put structures in clearer context.

We have included a new Figure 1 —figure supplement 2, which displays the suggested circle-representation of residues for previously unobserved folds. In addition, we have added more labels to Figure 1.

21. top of page 3 there is a. missing

We have inserted the full stop.

GGT is referred to but it has not been defined or previously mentioned

GGT (globular glial tauopathy) is defined at the top of page 2.

22. Some of the structures are missing, such as 46, which is important since it is an RNA structure.

Figure 1 only contains those that were solved to sufficient resolution to allow de novo atomic modelling with confidence. All structures solved are shown as XY cross-sections in Figure 1 —figure supplement 1. The RNA structure was not solved to sufficient resolution, and is shown in Figure 1 —figure supplement 1.

23. There appears to be a mistake when structure 11a is referred to as being in KCl and with C3 symmetry.

This has been corrected to 10a.